# Vector bionomics and vectorial capacity as emergent properties of mosquito behaviors and ecology

**Sean L. Wu**[1], **Héctor M. Sánchez C.**[1,2], **John M. Henry**[3], **Daniel T. Citron**[3], **Qian Zhang**[3], **Kelly Compton**[3], **Biyonka Liang**[1], **Amit Verma**[4], **Derek A. T. Cummings**[5], **Arnaud Le Menach**[6], **Thomas W. Scott**[7], **Anne L. Wilson**[8], **Steven W. Lindsay**[9], **Catherine L. Moyes**[10], **Penny A. Hancock**[10], **Tanya L. Russell**[11], **Thomas R. Burkot**[11], **John M. Marshall**[1], **Samson Kiware**[12], **Robert C. Reiner Jr**[3,13], **David L. Smith**[3,13]*

1 Divisions of Biostatistics & Epidemiology, University of California, Berkeley, Berkeley, California, United States of America, 2 Instituto Tecnológico y de Estudios Superiores de Monterrey, Monterrey, Mexico, 3 Institute for Health Metrics and Evaluation, University of Washington, Seattle, Washington, United States of America, 4 Emory University, Atlanta, Georgia, United States of America, 5 Department of Biology, University of Florida, Gainesville, Florida, United States of America, 6 Clinton Health Access Initiative, Boston, Massachusetts, United States of America, 7 University of California, Davis, California, United States of America, 8 Liverpool School of Tropical Tropical Medicine, Liverpool, United Kingdom, 9 Department of Biosciences, University of Durham, Durham, United Kingdom, 10 Big Data Institute, University of Oxford, Oxford, United Kingdom, 11 Australian Institute of Tropical Health and Medicine, James Cook University, Cairns, Australia, 12 Ifakara Health Institute, Environmental Health and Ecological Sciences Thematic Group, Ifakara, Tanzania, 13 Department of Health Metrics Sciences, School of Medicine, University of Washington, Seattle, Washington, United States of America

* smitdave@gmail.com

**Data Availability Statement:** The core code is in a GitHub release: https://github.com/dd-harp/MBITES/tree/v1.0/MBITES. Code for this manuscript is available in a companion GitHub

## Abstract

Mosquitoes are important vectors for pathogens that infect humans and other vertebrate animals. Some aspects of adult mosquito behavior and mosquito ecology play an important role in determining the capacity of vector populations to transmit pathogens. Here, we re-examine factors affecting the transmission of pathogens by mosquitoes using a new approach. Unlike most previous models, this framework considers the behavioral states and state transitions of adult mosquitoes through a sequence of activity bouts. We developed a new framework for individual-based simulation models called MBITES (Mosquito Bout-based and Individual-based Transmission Ecology Simulator). In MBITES, it is possible to build models that simulate the behavior and ecology of adult mosquitoes in exquisite detail on complex resource landscapes generated by spatial point processes. We also developed an ordinary differential equation model which is the Kolmogorov forward equations for models developed in MBITES under a specific set of simplifying assumptions. While mosquito infection and pathogen development are one possible part of a mosquito's state, that is not our main focus. Using extensive simulation using some models developed in MBITES, we show that vectorial capacity can be understood as an emergent property of simple behavioral algorithms interacting with complex resource landscapes, and that relative density or sparsity of resources and the need to search can have profound consequences for mosquito populations' capacity to transmit pathogens.

release: https://github.com/dd-harp/MBITES/tree/v1.0/scripts. Data produced by the computational experiments has been archived, and it is available at https://doi.org/10.6084/m9.figshare.12049002.

**Funding:** This work was funded by the Bill and Melinda Gates Foundation OPP1110495 (SW, HMSC, JH, QZ, DC, KC, RCR, DLS) and US National Institutes of Allergies and Infectious Diseases U19-A1089673 (DLS). SW and HMSC acknowledge support from the Malaria Elimination Initiative at University of California, San Francisco and the Parker Foundation. The funders played no role in any phase of this manuscript.

**Competing interests:** The authors have declared that no competing interests exist.

## Author summary

Mathematical modelling of pathogen transmission by mosquitoes began over a century ago with Ronald Ross and has produced a set of metrics that are the basis of measuring transmission. One crucial metric is vectorial capacity (VC), a simple equation describing the potential of mosquitoes to transmit pathogens. Despite its elegance, this formula lacks specificity to describe mosquitoes in a particular landscape. To study how these metrics arise in particular places, we built MBITES (Mosquito Bout-based and Individual-based Transmission Ecology Simulator), a complex stochastic individual-based model where mosquitoes fly from place to place to blood feed, sugar feed, lay eggs, mate, or rest. We also built a related model, MBDETES based on deterministic mathematics to show how the complex behaviors possible in MBITES can be summarized on average, and to provide a bridge to the simple equations describing VC. Through a series of computational experiments, we show a strong dependence VC and other metrics on fine details of the landscape mosquitoes inhabit which are not obvious from simple equations.

## Introduction

Mosquitoes transmit the pathogens that cause malaria, filariasis, dengue, and other diseases that account for approximately 17% of the global burden of infectious diseases [1]. Mosquito-borne pathogens are transmitted to a vertebrate host while mosquitoes probe or blood feed, so the intensity of transmission and the risk of infection are related to mosquito blood feeding behaviors and local population density. Development of concepts and metrics to measure transmission intensity by mosquitoes, such as the entomological inoculation rate (EIR) and vectorial capacity (VC), were motivated by or derived from mathematical models of pathogen transmission by mosquitoes [2–6]. Entomologists meanwhile identified and developed field methods to measure some of the parameters that are key determinants of the EIR and VC: mosquito survival, mosquito population density, the overall blood feeding frequency, and human blood index, the ratio of mosquitoes to humans in the area, and the pathogen's extrinsic incubation period (EIP) [7, 8]. These parameters are also important determinants of vector population responses to vector control interventions, such as insecticide-treated nets, indoor residual spraying, and spatial repellents, to name but a few. These parameters and metrics arise from basic mosquito behavioral algorithms for finding and using resources, avoiding hazards, and adjusting to weather and other factors that characterize mosquito ecology [9, 10]. An important question is how parameters relevant for pathogen transmission and vector control are co-determined by basic behavioral algorithms and genetically determined preferences of each vector species and by the availability and distribution of resources and other ecological factors.

The canonical model of the entomological aspects of malaria transmission, called the Ross-Macdonald model, describes changes in the sporozoite rate [4]. The associated formulae for VC and EIR, which arise from a common conceptual and mathematical basis, summarize transmission intensity in terms of a few parameters (Table 1) [6, 11]. Both formulae count the expected number of events occurring on one human on one day: VC measures the number of infectious bites that would arise from all mosquitoes blood feeding on a typical human, as if that human were perfectly infectious; and the daily EIR measures the number of infectious bites received by a typical human. The number of bites arising must approximately balance the number of bites received, after accounting for time lags (e.g. the EIP), mosquito migration, and imperfect transmission from infected humans. Mathematical formulas for VC and the

**Table 1. Vectorial capacity.** The five parameters comprising the classical formula for vectorial capacity (VC or denoted $V$), describing the total number of infectious bites arising from all the mosquitoes feeding on a single human on a single day under the assumptions of the Ross-Macdonald model [4, 6, 11]. The expected number of blood meals on the pathogen's hosts, summed over a mosquito lifespan, is given by the term $S = fQ/g$. The probability of surviving the EIP is $P = e^{-gn}$. Mosquito population density is $m = \lambda/g$. Under the assumptions, the formula for VC is $V = \lambda S^2 P$. In the Ross-Macdonald model, the entomological inoculation rate, $\mathcal{E}$, is related to VC by a formula: $\mathcal{E} = V\kappa/(1 + S\kappa) \approx V\kappa$, where $\kappa$ is the proportion of bites on humans that infects a mosquito; the approximation holds when $\kappa$ is small, such that mosquito super-infection is rare. In MBITES, the same quantity can be computed directly by Monte Carlo simulation.

| | |
|---|---|
| $m$ | The number of female mosquitoes per human |
| $f$ | The blood feeding rate, per mosquito, per day |
| $Q$ | Proportion of blood meals on the pathogens' hosts |
| $g$ | The mosquito death rate, per mosquito, per day |
| $n$ | Extrinsic Incubation Period (EIP), in days |
| $\lambda = gm$ | The number of female mosquitoes emerging, per human, per day |
| $S = fQ/g$ | The expected number of human blood meals per mosquito |
| $P = e^{-gn}$ | The probability of surviving the EIP |

EIR derived *a priori* are consequently related by a simple formula [12]. The quantitative logic supporting these metrics is both parsimonious and compelling, and these metrics and associated bionomic parameters form the basis of medical entomology and most models of mosquito-borne pathogen transmission [13].

While the Ross-Macdonald model and associated bionomic parameters are a useful way of summarizing overall transmission intensity, a weakness of the model is that it does not include many other parameters and metrics that are important for pathogen transmission dynamics, the measurement of transmission, and responses to vector control. These include metrics for heterogeneous biting by mosquitoes [14], the spatial dimensions of transmission or control [15–17], vector contact rates with and quantitative responses to various kinds of vector control deployed in myriad combinations and coverage levels [18], environmental factors that affect variance in the number of mosquitoes caught [19], and nuances of behavior affecting the accuracy of various field methods [8]. Models describing effect sizes of vector control rely on assumptions about the way interventions alter the basic bionomic parameters [20], but there has been very little theory developed to understand either what contextual factors determine baseline bionomic parameters or how contextual factors influence the effect sizes of control [21]. What is needed is a synthetic framework for weighing entomological heterogeneity: spatial and temporal heterogeneity in the availability of hosts and the rates of blood feeding on hosts; heterogeneity in the availability of habitats, egg laying, and mosquito population dynamics; age-specific mosquito mortality; mating and the availability of mates; energetics, sugar feeding and the availability of sugar sources; and the risks and costs associated with searching for all these resources. Understanding and quantifying the inter-dependency of mosquito behavior on hazards and resources through observation presents enormous challenges.

One way to forge a new synthesis is to model mosquito behavior at its most irreducible level and in extreme detail in order to prioritize new research. To that end, we here present a new framework for building individual-based models based on the concept of an "activity bout". A bout is initiated when a mosquito launches itself in the air to do something and ends after a mosquito has landed, rested, and is about to launch itself into the air again. It includes a sequence of events which may be of varying duration depending on factors both internal and external to the mosquito. In the models we present here, a bout is the irreducible unit of mosquito activity. Specific actions during the bout depend on the mosquito's current behavioral

state. This behavioral state depends on cues from local ecology and the mosquito's current physiological state and determines what activity the mosquito is intent on accomplishing at any given time, such as seeking to blood feed or oviposit. Success and survival through each bout depend on context and chance. Probabilities are affected by the resources and hazards in its vicinity, the cues it uses to find those resources, its efficiency in using those resources, and other factors such as vector interventions that suppress transmission by killing mosquitoes or altering their behavior. This description of behavioral states makes clear the joint dependence of behavioral state transitions on the mosquito's biology as well as local ecology.

To understand how ecology and behavior jointly affect transmission, we developed bout-based behavioral state models for mosquitoes to show how the values of bionomic parameters in the Ross-Macdonald model arise from basic mosquito behavioral algorithms in response to ecology. These models consider a behavioral state space and model the dynamic transitions between states as they follow the biological imperatives of their state and succeed or fail depending on the availability of local resources and other factors. MBITES (Mosquito Bout-based and Individual-based Transmission Ecology Simulator) is a framework for building individual-based simulation models of mosquito behavioral activity bouts in exquisite detail. Simulated mosquito activities are implemented as algorithms executing activity bouts and behavioral transitions in response to resources that are organized on spatially explicit land-scapes. MBDETES (Mosquito Bout-based Differential Equation-based Transmission Ecology Simulator) is a differential equation based model of behavioral state transitions that is more analytically tractable than MBITES. Under a restricted set of assumptions, some MBITES models can be represented as a continuous-time Markov process, for which MBDETES is the set of master equations. Because MBDETES describes the expected behavior of some MBITES models, we use the two frameworks for mutual verification through both simulation and analysis.

These two bout-based behavioral state models make it possible to investigate how behavioral algorithms and resource distributions affect local decisions and give rise to the parameters that are widely acknowledged to be important for pathogen transmission. Through Monte Carlo simulation of models developed in MBITES, it is possible to compute any quantity describing adult mosquito reproductive success or capability to act as effective pathogen vectors and, through careful *in silico* analysis, to learn what factors determine their values. To this end, MBITES provides algorithms to compute from simulation output: lifespans, metrics of mosquito dispersal from the natal aquatic habitat, stability index (number of human blood meals per mosquito lifetime), the length of a feeding cycle, survival through one feeding cycle, blood feeding rate, entomological inoculation rate (EIR), egg production and dispersal, vectorial capacity (VC), and the spatial scales over with VC is dispersed. By summarizing models according to these values, we can map these behavioral models to regularly discussed and estimated metrics that are the target of inference and control in many intervention studies. We show how the classical mosquito bionomic parameters used by the Ross-Macdonald model arise from mosquito behavioral algorithms in their ecological context, but we also describe their distribution and spatial dimensions. These behavioral state models thus provide a way of synthesizing more than a century of studies that have observed and measured aspects of individual mosquito behavior in a variety of contexts, from laboratory through the field.

## Methods

### MBITES

MBITES is a framework for building individual-based continuous-time discrete-event simulation models for adult mosquito behavior and ecology. The framework is highly mimetic:

simulated mosquito activity is designed to resemble what we believe actual mosquitoes are doing. The descriptions of mosquito behavior in the sections below map onto the structures and algorithms that are built into MBITES. For each simulated mosquito, MBITES samples events and outputs a lifetime trajectory through behavioral state space as well as spatial location. The framework accommodates differences in behavioral algorithms and life-history traits across species, different ecological contexts, and different purposes. To make it useful as a research tool, some pre-defined behavioral states are optional. An important feature of MBITES is a resource landscape can be configured to suit any situation. MBITES has a modular design: several functions can be called to model each biological or ecological process, and it is comparatively easy to add, remove or modify new functions or features. Mathematical and computational details, including functional forms and some options are documented in the code.

MBITES is under active development and it is being maintained with version control. This document describes MBITES version 1.0. The code was written in R and C [22]. Source code for MBITES are released at a permanent GitHub archive (https://github.com/dd-harp/MBITES/tree/v1.0/MBITES); configuration files for simulations are also available here (https://github.com/dd-harp/MBITES/tree/v1.0/scripts).

**The landscape.**   MBITES simulates mosquito behavior on a set of points in space called *haunts*, small areas where mosquitoes can rest between long range search flights. *Resources* may be present at some of these haunts. Landing spots within a *haunt* are represented as a set of micro-sites. The set of points and associated resources is called the *landscape*.

Each adult mosquito emerges from one of these haunts and moves among the haunts throughout its life as it searches for and utilizes the resources it needs. In this manuscript we consider only two types of resources. First, blood feeding haunts are places where vertebrate animals are typically found such that mosquitoes could take a blood meal, which could include the area around a family dwelling or other structure, a field where farming occurs, or an outdoor spot where humans or other vertebrate animals are found. Second, some haunts include aquatic *habitats* where mosquitoes attempt to lay eggs. In this manuscript we do not consider density-dependent dynamics in aquatic habitats, as our focus is on the behavioral algorithms that structure how cohorts of mosquitoes behave after emergence. We acknowledge the possibility and importance of nonlinear dependence in adult mosquito behaviors, such as the choice of where to oviposit, and competition for resources by larval mosquitoes in aquatic habitats. However, we feel confident in presenting the current set of *adult* behavioral algorithms as an initial step, to be followed up with development of detailed *aquatic* algorithms which will incorporate such dynamics; the modular design patterns we have used in MBITES allows for such development to be feasible and transparent.

Two other types of resources included in MBITES, which will be described in detail in future manuscripts, are sugar sources and mating sites. Each haunt may have one or more resource type and more than one of each type of resource; more than one potential blood host could be present, and there could be more than one suitable aquatic habitat, such as a pond and a rain-storage barrel in the backyard of a home, for instance. The question of how to model a landscape—how many haunts make up the landscape, how large an area is represented by each haunt, whether to include haunts without other resources, and how many micro-sites in each haunt—is flexible. In some instances, a haunt could represent a small area around a habitat or the area immediately surrounding a single household. The question is intrinsically related to mosquito search and dispersal, a question that may require some tuning, depending on the purpose of a simulation.

In addition to the resources present, each haunt is assigned a set of *local hazards*, or parameters that affect survival at the site, which reflects highly local conditions (such as predators) that may make a haunt more or less dangerous than others of the same type.

Each haunt is also characterized by a set of micro-sites, which define specific aspects of the places within a haunt where the mosquito lands to rest. Different *types* of haunts may have different sets of micro-sites with different resting surfaces. A *homestead* is a pre-defined haunt type with three "micro-sites" a mosquito may rest either in the house, or outside on the house, or outside on surrounding vegetation. If there is no human dwelling at a haunt, it could have only vegetation. Other types can be easily constructed (*e.g.*, a homestead with livestock sheds) with their own set of relevant micro-sites for resting.

These micro-sites were devised to simulate survival through the post-prandial resting period or contact with various kinds of vector control in a highly realistic way, including exposure to residual pesticides and house entering. Housing quality and housing improvements can affect the probability of entering a house; when a mosquito attempts to enter a house, it may encounter an eave tube; insecticide spraying can be applied to the interior walls of houses (or not), their exterior walls (or not), or vegetation (or not) so that contact is simulated only if a mosquito lands on the type of micro-site that has been sprayed; and area repellents or other local features can make it more likely a mosquito will leave the haunt, thereby initiating a new search bout. Encounters with environmental vector based intervention, such as indoor residual spraying (excluding personal-level interventions, such as topical repellents, which we considered separately) at any point may incur death or physiological damage.

**Modular design.**  MBITES is designed to be nested within a broader framework for simulating the transmission dynamics and control of mosquito-borne pathogens. Because individual mosquitoes are simulated as a continuous-time discrete-event process, a mosquito's actions can be simulated exactly and do not need to be discretized to the nearest time step. While each mosquito agent is simulated exactly, between-agent synchronization occurs at fixed time steps. Synchronization allows agents to update each other on where they are, how many resources have been consumed, etc. in order to simulate interaction. In all simulations for this paper we chose the synchronization time step as one day. In principle, any synchronization time step can be used so long as it is short enough that the consequences of one mosquito (or agent) are synchronized before they would affect another.

Because agents in MBITES interact on the landscape, it is necessary for haunts to have associated data structures that record information to pass between different parts of the model. For example, when a human visits a certain blood feeding haunt, they must leave a piece of information denoting their id, when they arrived, and the duration of their stay so that during the mosquito portion of the MBITES simulation, mosquitoes visiting that haunt have a list of potential blood hosts they can select from to take a blood meal. We call these data structures *queues*, and they allow different modules to interact in a generic way, facilitating a design that is both flexible and extensible. To continue the blood feeding example, it is not important for mosquitoes to know the specific algorithms by which human movement between blood feeding haunts is simulated, as long as when a mosquito arrives at a haunt to take a blood meal, it can query the queue there to find out which blood hosts are available. It is information contained in these queues that are synchronized in each time step.

As described, at each blood feeding haunt, MBITES tracks all available blood meal hosts in a queue called the *atRiskQ*. Because not all hosts present at a haunt are equally available or attractive to the mosquito, the atRiskQ stores not only the identities of blood hosts but also a biting weight for each human and other potential blood host. These biting weights reflect a combination of time spent and comparative attractiveness of all the hosts. When a mosquito

chooses a blood host, she samples from the discrete distribution on blood hosts parameterized by those biting weights.

Each haunt also has a queue object called the *eggQ* that lists aquatic habitats at the haunt and ovitraps. The eggQ stores any eggs laid by mosquitoes for potential use by linked models of immature mosquito population dynamics in aquatic habitats. Like potential blood hosts, these habitats are not equally available or attractive to the mosquito, so each eggQ has a search weight. These weights can be changed dynamically to mimic specific behaviors; for example, it is possible to change the search weight if larvae are present [23].

A landscape is thus comprised of the locations of multiple points in space called haunts where mosquitoes rest, their resources, a set of micro-sites describing landing spots (*e.g.*, homestead, shed, field, forest edge), local hazards, and queues. Importantly, the ability to characterize landing spots in varying granular levels of detail allows for specific simulations that can adapt to levels of detail present in field data. In this way, complex composite types of haunts can be built up by adding the basic elements, as in our previous example of a single blood feeding haunt with several independent aquatic habitats representing a large house with multiple mosquito breeding ponds in the backyard; the flexibility can accommodate complex ecological dynamics of structured habitat and behavior such as skip oviposition. All these elements determine where a mosquito attempts to do something on the landscape and affects its probability of success and survival.

**Mosquito dispersal.**   Mosquito movement in MBITES may occur during a behavioral bout if a *search* is triggered. It is a relocation from one haunt to a different haunt. A mosquito may decide to leave a haunt for several reasons; if the mosquito survives the search flight, it picks a destination according to a probability mass function, which we call the haunt-specific movement kernel. While movement is random in the sense that it is sampled from this distribution, the form of this distribution is not constrained. Diffusive movement could be approximated by a landscape sufficiently rich in haunts where the haunt-specific movement kernel followed simple nearest neighbor rules, but other models are also possible.

The decision of how to configure haunts and how to model movement is not determined within MBITES. A user chooses the number of haunts and their locations knowing these are the only locations where a mosquito can be. Computation of the probability distribution functions describing dispersal to each haunts from any starting point is done prior to the MBITES simulation; the simulation accepts as input the computed probability vectors giving interpoint movement. Haunts could thus include a set of points describing all areas where mosquitoes rest or only those that are significant because of the presence of other resources. Since dispersal is modeled pre-processing, the user has complete control over the proportion that survive dispersal and the probability of reaching each one of the other haunts.

As utilities for setting up simulations, MBITES has developed some simple parametric functions to compute these probabilities as a function of distance and an activity-specific *search weight*, $\omega_x$, where $x$ denotes a particular blood feeding haunt or aquatic habitat [14]. These functions thus provide multiple ways of computing the probability of moving from one haunt to any other haunt, optionally conditional on behavioral state.

**Behavioral states and other variables.**   Simulations in MBITES include variables describing mosquito behavioral states and other states that maybe relevant for survival [8]. At any point during a mosquito's lifespan, it will seek to mate, sugar feed, blood feed, or lay eggs; these are biological imperatives that must be accomplished to survive and reproduce. In terms of simulation, this means that at all points in time a mosquito is alive, it belongs to one of a discrete set of behavioral states that govern its actions while in that state, as well as what state it is likely to transition to next.

Transitions between behavioral states were developed around a basic description of adult mosquito behavior. A newly emerged female mosquito must harden and mate before it is mature, whereupon it begins a cycle of blood feeding and egg laying throughout the rest of its life. After a search to find suitable blood meal hosts, the mosquito selects a host and approaches it in an attempt to blood feed. Assuming it is successful, after the blood meal, the mosquito typically rests to lose some of the water weight in a post-prandial resting period. During this time, blood is provisioned into eggs that require some time to mature. Once the egg batch is mature, the mosquito initiates a search to find a suitable aquatic habitat and then lays eggs. Sugar feeding occurs frequently throughout a mosquito's life, depending on availability of resources and energy levels, and both sexes participate in the activity [24].

In order to allow detailed simulation of activities mosquitoes undertake during each behavioral state, as well as allowing for flexibility to account for differences among species, behavioral algorithms simulate specific actions mosquitoes take as they attempt to accomplish goals associated with each state, as well as transitions to future states. While there is a general pattern to be followed (*e.g.* blood feed, rest, lay eggs, and repeat), each mosquito follows a probabilistic sequence through a set of distinct phases surrounding the activity bout as they seek to accomplish their goals. Transitions between behavioral states depend on various internal characteristics of the individual mosquito and logical prerequisites. Egg batches must mature before a mosquito is considered gravid and enters the egg laying state. If a mosquito is gravid, it will tend to lay eggs, though re-feeding can occur regardless (see below). Otherwise, disregarding sugar feeding and mating for the moment, a mosquito's state is oriented towards blood feeding. A mosquito must leave a haunt to initiate a search if the resources it needs are not present, but a decision to leave the haunt and initiate a search can also occur even if resources are present, depending on other events that occur during a bout and properties of a haunt. For example, area repellents could increase the probability of a failure, and force a mosquito to initiate a new search.

In addition to a mosquito's behavioral states, MBITES includes a set of other variables making it possible to model heterogeneity among individual mosquitoes in extreme detail. Each mosquito in MBITES is described by internal variables that include but are not limited to: physical and physiological condition, energy reserves, size of the most recent blood meal, the number of mature eggs ready to be laid, infection status, physical condition, and a set of variables related to sugar feeding and mating. For example, inter-site movement is physiologically stressful on the mosquito, and during travel between haunts, a mosquito's energy level (an optional variable) can be decremented. The energy level can be replenished by blood meals (for females), or sugar feeding, and could be modeled as an important source of mortality while searching in resource-sparse environments. A random variate is drawn to determine the amount of physical damage (*e.g.*, wing tattering) that was incurred and modifies its physical condition or physiological damage (*e.g.*, after exposure to insecticides). Physical and physiological damage takes a cumulative toll on the mosquito.

MBITES is capable of modeling both female and male mosquito populations and behavior. For female mosquitoes, the primary behavioral states are blood feeding and egg laying (the two necessary components of the gonotrophic cycle). In this manuscript, male mosquitoes are not considered and the optional behavioral states of sugar feeding and mating and all associated variables have been turned off in order to introduce and focus on algorithms describing blood feeding and egg laying by females.

Each behavioral state—blood feeding or egg laying—requires one or more activity bouts. Additionally, each behavioral state may require one or more *types* of activity, namely *searching*

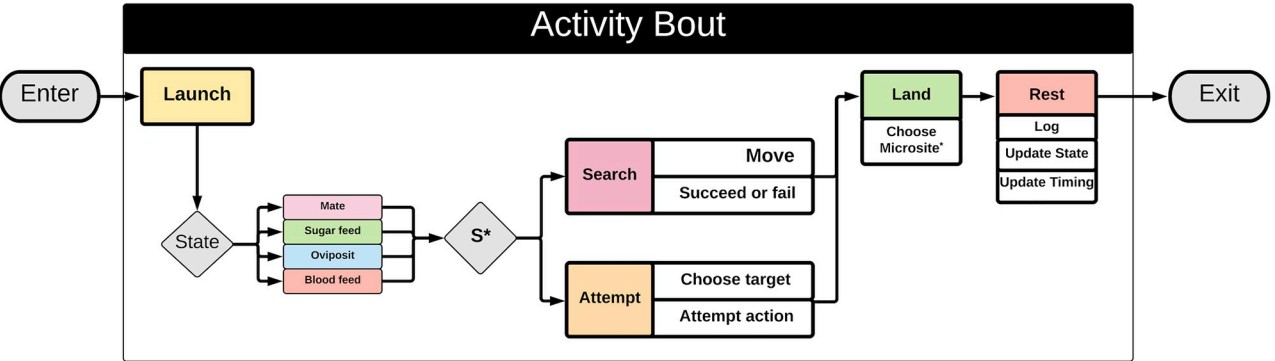

**Fig 1. Structure of an activity bout. Top)** MBITES and MBDETES model mosquito behavioral states and state transitions required for the gonotrophic cycle. The first two columns list the behavioral states, and the last two columns describe the potential state transitions. A mosquito is either searching for a blood host (F) or attempting to blood feed (B), searching for aquatic habitat (L) or attempting to oviposit (O), or resting (R). Transitions depend on whether the last bout was a success or failure, and optionally on refeeding behavior† or laying a partial egg batch and skip oviposit‡. The next activity bout is also affected by whether a mosquito decides to make an attempt or initiate a search*. **Bottom)** In MBITES models, each behavioral state has an associated activity bout that has a common structure, as illustrated in the diagram. The activity bout involves a sequence of four phases: launch, do an activity (either a search or an attempt), land, and rest. The type of activity is determined both by its behavioral state and by presence and availability of resources. A mosquito will stay (S) unless there are no resources present or if the mosquito has become frustrated (*), in which case it will initiate a search. If the mosquito decides to stay, it makes a choice and an approach that may or may not succeed at what it was trying to do. When a mosquito lands, it selects a micro-site for a resting spot from the set of possibilities at that site. During the resting period, data from the last bout are logged, the behavioral state is updated, and the waiting time to the next launch is determined. A mosquito enters the bout either after emerging from aquatic habitat or after exiting its previous bout and surviving.

or *attempting*. Thus the set of possible activity bout types each associated with a behavioral state has been denoted by a letter (Fig 1, Table 2): the blood feeding search bout (F); the blood feeding attempt bout (B); the search bout for egg laying (L); the egg laying attempt bout (O). A letter is also assigned to the post-prandial resting period (R) that always follows a blood meal,

**Table 2. State transitions & waiting times.** In MBITES, it is possible to compute the expected state transitions and waiting times from any state to the next state. In MBDETES and limiting cases of MBITES, these single-state transition expectations can be used to estimate the state transition probabilities and waiting times from one state to every other state, including the length of a gonotrophic cycle, from resting to resting (*i.e.*, from $R \to R$). The table gives formulas for the probability of surviving to reach the behavioral state $Y$ starting from another state $X$, $\Psi_{X,Y}$, where $X, Y \in \{L, O, F, B, R\}$. Note that $P_{X,Y}$ denotes the single activity bout probability of a state transition. It also gives the expected waiting time to $Y$ from $X$ is $T_{X,Y}$. These formulae are expressed in terms of the single bout state transitions and waiting times, $P_{X,Y}$ and $T_X = \gamma_X^{-1}$ (or they can be by making a simple substitution from one of the formulas appearing in the table above it).

| | $\Psi_{X,Y}$ | $T_{X,Y}$ |
|---|---|---|
| $L \to O$ | $\frac{P_{L,O}}{1-P_{L,L}}$ | $\frac{T_L}{1-P_{L,L}}$ |
| $O \to F$ | $\frac{P_{O,F}}{1-P_{O,O}-P_{O,L}\Psi_{L,O}}$ | $\frac{T_O+P_{O,L}\Psi_{L,O}T_{L,O}}{1-P_{O,O}-P_{O,L}\Psi_{L,O}}$ |
| $O \to B$ | $\frac{P_{O,B}}{1-P_{O,O}-P_{O,L}\Psi_{L,O}}$ | $\frac{T_O+P_{O,L}\Psi_{L,O}T_{L,O}}{1-P_{O,O}-P_{O,L}\Psi_{L,O}}$ |
| $F \to B$ | $\frac{P_{F,B}}{1-P_{F,F}}$ | $\frac{T_F}{1-P_{F,F}}$ |
| $B \to R$ | $\frac{P_{B,R}}{1-P_{B,B}-P_{B,F}\Psi_{B,F}}$ | $\frac{T_B+P_{B,F}\Psi_{F,B}T_{F,B}}{1-P_{B,B}-P_{B,F}\Psi_{F,B}}$ |
| $F \to R$ | $\Psi_{F,B}\Psi_{B,R}$ | $T_{F,B} + T_{B,R}$ |
| $O \to R$ | $\Psi_{O,F}\Psi_{F,R} + \Psi_{O,B}\Psi_{B,R}$ | $\frac{P_{O,F}}{P_{O,F}+P_{O,B}}\left(T_{O,F} + T_{F,R}\right) + \frac{P_{O,B}}{P_{O,F}+P_{O,B}}\left(T_{O,B} + T_{B,R}\right)$ |
| $L \to R$ | $\Psi_{L,O}\Psi_{O,R}$ | $T_{L,O} + T_{O,R}$ |
| $R \to R$ | $\sum_{X \neq R} P_{R,X}\Psi_{X,R}$ | $T_R + \sum_{X \neq R} P_{R,X} T_{X,R}$ |

which is a part of the blood feeding attempt bout. If a mosquito dies, it's behavioral state is set to (D).

| State / Activity | Name | Success | Fail or Repeat† |
|---|---|---|---|
| Blood Feeding Search Bout | **F** | **B** | **F** |
| Blood Feeding Attempt Bout | **B** | **R** | **B\|F**∗ |
| Post Prandial Resting Period | **R** | **L\|O**∗ | **B**† |
| Egg Laying Search Bout | **L** | **O** | **L** |
| Egg Laying Attempt Bout | **O** | **B\|F**∗ | **L\|O**‡ |

**The activity bout.** Regardless of behavioral state, all activity bouts have four phases: launch, do an activity, land, and rest (Fig 1). Launching itself into the air, finding a suitable landing spot, and resting prior to the next launch must occur regardless of the biological state of the mosquito (blood feeding or oviposition, in this manuscript) and are a common part of each behavioral state and activity bout. However because the purpose of activity bouts is to accomplish different biological goals, the activity phase of each bout depends on the behavioral state. The specific activities undertaken differ depending on behavioral state, and it is possible to modify these activities to describe what happens during each bout in virtually unlimited detail. The endpoint of each bout is either death or another bout and possibly a behavioral state transition (Table 2, Fig 1).

The paths through a bout are sampled from appropriate distributions that determine whether the bout resulted in death, success (state transition), or failure (remain in the same behavioral state). These activity bouts, prefaced by a launch and ending with a rest, may be of varying duration, but encompasses all of the activity by a mosquito from launch to launch.

The modular nature of MBITES makes it possible to configure all these options to consider a biological process of interest. While all bouts share similar structure and call on similar functions (*e.g.* optionally, flights could expend energy or contribute to cumulative wing damage), the activities and outcomes will differ based on each individual mosquito's internal physiological state. Thus, a mosquito's life consists of a series of transitions between behavioral states, each of which may take several bouts to accomplish, completely determining a mosquito's activity throughout its life.

**Launch and timing.** A new bout begins the moment a mosquito launches itself into the air. The timing of the start of the launch phase is determined during the previous bout during the resting phase following landing. By sampling the time to next launch at this point in the simulation, launch times can be (optionally) conditional on events that have taken place during the previous bout.

**Do an activity—Search or attempt.** After launching, the specific behavioral algorithms called by the mosquito depend on its behavioral state (decision point "state" in Fig 1), as well as the local distribution of resources necessary to fulfill the mosquito's current biological needs. A mosquito will either "attempt" to accomplish the task required by their behavioral state or "search" for the resource it needs to accomplish that task. In any particular activity bout, a mosquito will either search or attempt, but not both. Searching or attempting algorithms are behavioral state-specific, and will be called during that phase of the bout (attempting to blood feed, as illustrated in Fig 2, for example, means something very different behaviorally than attempting to oviposit, as illustrated in Fig 3). Details of the attempt algorithms are described below.

# Blood Feeding Attempt Bout

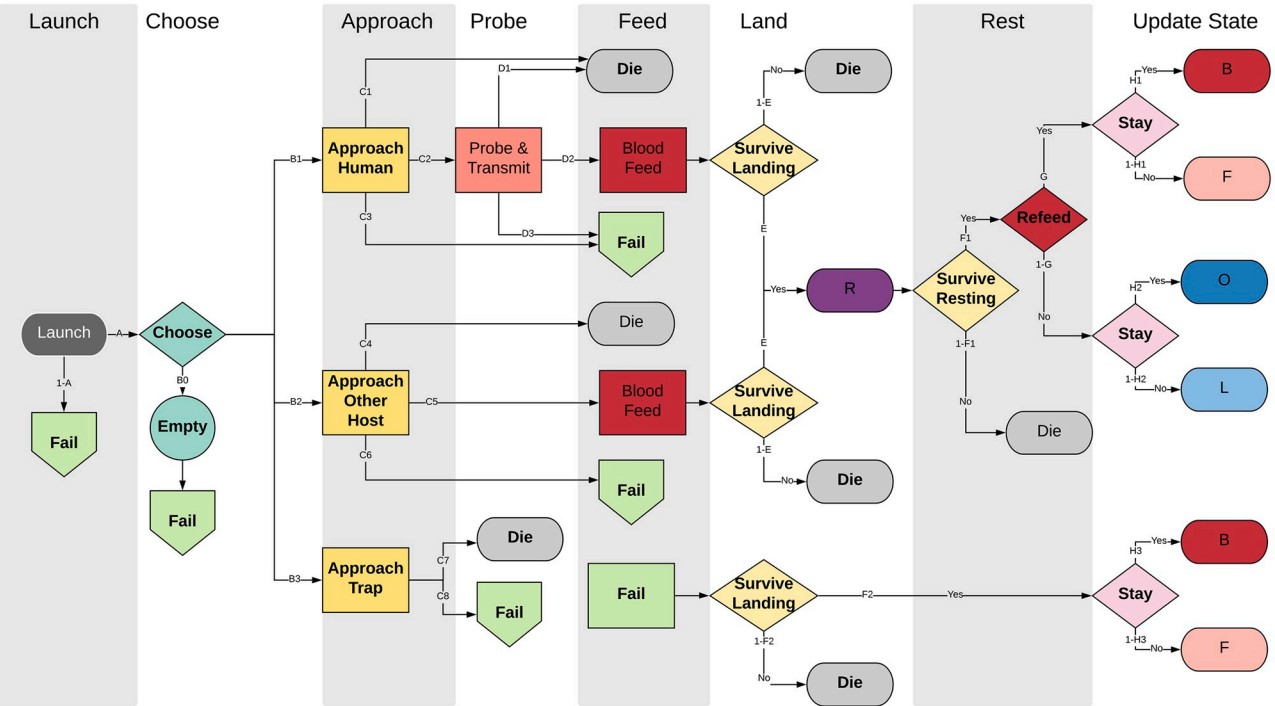

**Fig 2. The structure of a blood feeding attempt bout in MBITES.** The flowchart follows the progression of a mosquito through simulated events, from the launch (dark grey oval), choosing a host from the atRiskQ (aquamarine diamond), and the events that follow depending on what sort of host was chosen (yellow rectangles). If a human is chosen (or more generally, a blood host that is also a host for the pathogen), then each mosquito must approach and attempt to probe (salmon rectangle) and then blood feed (red rectangle). If a non-human host is, probing is ignored. Traps mimicking a blood host can also be chosen. After a blood meal (red rectangle), a mosquito must land and choose a resting spot (yellow diamonds). A post-prandial resting period follows a successful blood meal which has its own hazards (purple oval), including additional hazards associated with a flight laden with blood, which may be followed by decision to feed again (dark red diamond). Similarly, after failing the attempt (green pentagons to green rectangle), a mosquito must land and choose a resting spot (yellow diamonds). At each step, it is possible to die (light grey ovals). At any point when failure occurs or during landing, a mosquito could choose to leave the haunt and initiate a search on the next bout. This condition is checked after completing the bout (pink diamond). At the end of a bout, the mosquito's behavioral state and other state variables are updated. The endpoint of each bout is either death (grey ovals), a repeated blood feeding attempt (dark red oval) or a state transition to either a blood feeding search (pink oval) or to oviposit (blue ovals).

The searching algorithm moves a mosquito to a new haunt on the landscape to find resources, as described earlier in section Mosquito Dispersal, while the attempting algorithm describes how a mosquito fulfills its behavioral imperatives once necessary resources are present, which will be described below for egg laying and blood feeding.

There are many factors that determine whether a mosquito will make an attempt or begin a search. A mosquito will stay and make an attempt if the necessary resource is present at the haunt, but it could leave and initiate a search if it has previously been frustrated in its attempts or if the resource is not present. Many events occurring during the bout can trigger a decision to leave and initiate a search during the next bout. For example, if the mosquito is primed to blood feed and blood hosts are present, the mosquito will tend to approach a potential blood host and try to blood feed. If there are no blood hosts present, the mosquito will search and move to a new haunt. Even if blood hosts are present, after multiple failed attempts, or if the blood hosts are not sufficiently attractive, a mosquito may become frustrated and leave.

**Land.** After its flight, a mosquito must land and rest. This must occur at one of the micro-sites at a haunt, as described earlier. During the landing phase, an algorithm called *restingSpot*

# Egg Laying Attempt Bout

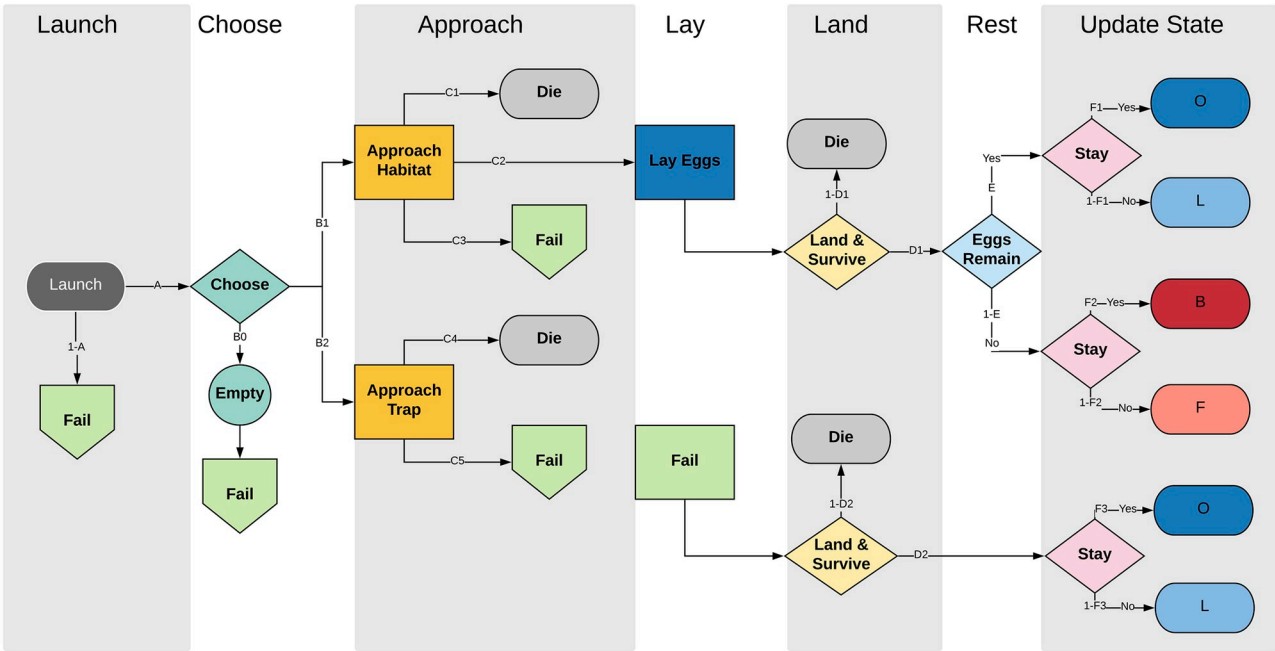

**Fig 3. The structure of an egg laying attempt bout in MBITES.** The flowchart follows the progression of a mosquito through simulated events, from the launch, choosing a habitat or trap from the eggQ (aquamarine diamond), and the events that follow depending on whether the mosquito chose a habitat or a trap (yellow rectangles). If a mosquito approaches the habitat, it could lay eggs. Alternatively, a mosquito could approach a trap and fail in the approach (thus surviving) or die (light grey ovals). If a mosquito is deterred in the approach to its habitat or the trap, it fails (green pentagons to green rectangle). After a successful approach to a habitat, a mosquito lays eggs (blue rectangle). After laying eggs or failing, a mosquito must land and survive (yellow diamonds). If not all eggs were laid, a mosquito can choose another habitat to lay (light blue diamond). At any point when failure occurs or during landing, a mosquito could choose to leave the haunt and initiate a search on the next bout. This condition is checked after completing the bout (pink diamonds). At the end of a bout, the mosquito's behavioral state and other state variables are updated. The outcome of each bout is either death (light grey ovals), a repeated egg laying attempt bout (dark blue ovals) or a state transition to either an egg laying search (light blue ovals) or a blood feeding attempt (red or pink ovals).

simulates the mosquitoes choice of micro-site, and the consequences of landing there (*e.g.* encountering a sprayed surface or entering or leaving a house). Movement to a micro-site within a haunt to land and rest is designed to emulate short hops (*e.g.*, <10m) a mosquito may take that differs from longer range searching behavior. The algorithm also tracks if the attempt was successful. Repeated failures at the same site may lead to the mosquito becoming frustrated, in which case the next activity bout will include a search regardless of resources availability.

**Rest.**    The duration of the resting phase, the time elapsed between landing and the next launch, will in general differ based on behavioral state, other state variables, and events that may have transpired during the bout. Several probability distribution functions functions are available to simulate heterogeneity in the time required by mosquitoes to accomplish certain tasks based on individual characteristics and spatial location. The default option is for these times to follow an exponential distribution.

During the resting phase, internal characteristics of the mosquito are updated, including cumulative wing damage, and internal energy reserves which have been used for flight. Egg batches (if present) are also checked for maturity, depending on if the mosquito has successfully blood fed, and if so how much. The next behavioral state is determined by the success or

failure of the current bout. Other variables are also updated during this time, such as age and (if a pathogen model is present), infection status.

**Survival.**   A mosquito can die at many points during a bout as a result of specific events, such as being swatted while attempting to blood feed, or as a result of contact with some vector control device. Survival through a bout is also computed during every resting phase. Mortality is associated with the stress of flight, which sets a baseline mortality probability (per flight). A mosquito must also survive any *local hazards*, which determine the site-specific probability of dying while landing or flying at each site, such as from predation by jumping spiders or dragon flies.

**Attempt algorithms.**   The following sections describe in detail the algorithms that determine how a mosquito makes an attempt to blood feed or oviposit. In addition, a separate section discusses the post-prandial resting period and algorithms for simulating oogenesis and re-feeding.

**Blood feeding attempt and pathogen transmission [B].**   During an activity bout in which a mosquito attempts to blood feed (Fig 2), a detailed sequence of events describing the blood feeding attempt is simulated, any of which can result in a failure (the green nodes in Fig 2) with its associated state transitions (the green box in lower right of Fig 2). In the event of failure, the mosquito could either fail to survive the landing, be frustrated and leave the site to search for other blood hosts (a blood feeding search, F), or remain and try again (B).

When the mosquito attempts to feed on a blood host after launching, a sequence of critical processes must be simulated. Upon arriving at a site with viable blood hosts, the mosquito chooses a particular host at that site to approach from the atRiskQ object at that site, which maintains a list of available hosts (described in the section Modular Design).

If the mosquito chooses a human host, a detailed sequence of events is simulated. First, the mosquito must approach and land on the host successfully. The approach can result in failure or death if it is deterred by swatting or a sudden movement or something else. If the mosquito survives the approach and lands on the human host, it will try to probe the host, which can also result in failure or death. Finally, if probing is successful, the mosquito proceeds to feed. When a mosquito blood feeds, a random variable is drawn to determine the blood meal size, which affects egg maturation and batch size. Notably, the blood meal could infect a mosquito if the human host is infectious. Following successful feeding, the mosquito will make a short hop within the current site and find a resting spot to digest the blood meal (post-prandial rest, purple oval), simulated by the *restingSpot* algorithm, where the specific micro-site for resting is chosen from the set of micro-sites available.

Note that it is during probing that mosquitoes could transmit pathogens from their salivary glands into humans infecting them. During blood feeding, mosquitoes could take up pathogens in blood and become infected. This functionality is part of MBITES, but the details are beyond the scope of this paper.

If the mosquito chooses a non-human host (or more generally, a blood host that is not suitable for the pathogen), probing is not modeled. A mosquito can either survive and blood feed, survive without blood feeding, or die. If the mosquito survives, restingSpot is called (as above).

**The post-prandial resting period and oogenesis [R].**   The postprandial resting period is part of the blood feeding attempt bout following landing (purple oval, Fig 2), but it requires special consideration. After calling restingSpot, a long delay is simulated which represents the time needed for diuresis and the early stages of digestion to occur. Several options are available for modeling survival and the events that follow as blood is concentrated and provisioned into eggs. First, mortality rates can be higher because the mosquito is heavy with blood, increasing the chance that they fall victim to successful predation [25]. Second, a function has been provided to model overfeeding: the physiological stress of processing a blood meal can kill a

mosquito, so it is possible to model mortality as an increasing function of blood meal size. A function determines how much, if at all, the energy reserves are topped up by the blood in the blood meal.

During the resting period, an egg batch is produced; however, the mosquito will not be considered *gravid* until the batch is mature. The number of eggs in a batch is determined by a probability distribution function family, such as a discretized Gaussian distribution. Oogenesis (production of an egg batch) and the time delay to egg maturation is modeled explicitly in MBITES. Oogenesis is modeled in three ways: first, a simple phenomenological model, which simply draws a random variate from a Gaussian distribution with mean and variance parameterized to match entomological understanding; second, a semi-mechanistic model in which the size of the egg batch is linearly proportional to the size of the blood meal, in which a new blood meal adds to the egg batch size; and third, a model in which the egg batch is linearly proportional to the size of the blood meal, but eggs from a new blood meal completely replace a previous egg batch. The third model is unrealistic but mathematically convenient (*i.e.*, it was included so that there would be a memoryless model that could exactly match the assumptions of MBDETES).

In the first oogenesis model, each egg in the egg batch is considered to require a certain provision of blood for its maturation. Because each egg requires its own resources, the total blood provision needed for a batch is a linear function of the egg batch size. After the blood provision has been fulfilled, which could require multiple blood meals, the egg batch is mature and the mosquito will become *gravid*.

In the other two models, time to maturation is modeled as a delay between the initial biological commitment of the mosquito to produce a batch of eggs, and the time when the mosquito is considered gravid, upon which it will be primed to transition to the egg laying behavioral state. The time to maturation implements a simple phenomenological model, where a random variate is drawn for the maturation time which is coincident with the resting period; after this time delay, the mosquito will become gravid and either search for suitable aquatic habitats (L) or attempt to oviposit (O), depending on the local availability of resources. Re-feeding can occur in both these models.

**Re-feeding.** Re-feeding (the red diamond in Fig 2) is possible. A mosquito may take multiple blood meals prior to oviposition, which lengthens the time interval between successive oviposition attempts. Re-feeding depends in part on the model for oogenesis, as an oogenesis model could explicitly force re-feeding when a batch of eggs is not yet mature.

When re-feeding is not forced by the oogenesis model, MBITES allows for probabilistic re-feeding behavior that can optionally be disabled. Re-feeding behavior is a Bernoulli event that occurs when a mosquito is exiting the post-prandial rest (R) at the end of a blood feeding attempt bout. The probability of re-feeding is a function of either the size of the previous blood meal or the previous egg batch, accounting for mosquito propensity to top up the size of the egg batch before ovipositing.

**Egg laying attempt and oviposition [O].** After arriving at a haunt containing an aquatic habitat (i.e. after a search), or if the mosquito is already at a suitable haunt with one or more aquatic habitats, the mosquito chooses a particular habitat, or possibly an ovitrap (Fig 3). If a mosquito chooses an aquatic habitat, it will approach the habitat and attempt to lay eggs. The outcome of an egg laying attempt bout is either death, a failed attempt, or oviposition. Like the blood feeding attempt bout, mosquitoes can fail at any time during the attempt (green pentagons), which, if the mosquito survives the failed attempt, will lead it to either reattempt at the current site (L), or become frustrated and search for a new site (O). If a mosquito is successful it lays some fraction of her eggs in an aquatic habitat. If an ovitrap exists at a site, it competes

for attractiveness with other aquatic habitats that may be present, and ovipositing mosquitoes could become trapped and die in it.

## MBDETES

MBDETES (Mosquito Bout-based Differential Equation Transmission Ecology Simulator) is a system of coupled ordinary differential equations for modeling mosquito bout-based behavioral states and state transitions that offers analytical tractability but reduced flexibility compared with MBITES. MBDETES was developed as a way of approximating some models developed in MBITES: any MBITES model in which the behavioral state transitions are memoryless and the waiting time to the next state transition is always exponentially distributed is the the stochastic analogue of a model in MBDETES.

The analytic tractability of MBDETES and speed of numerical solutions serves several purposes. First, we can check that the complex behavioral algorithms in MBITES are indeed functioning correctly, by comparing the output of Monte Carlo simulations against analytic solutions in situations where their expected behavior is known. Second, MBDETES provides a simple null case against which the importance of process stochasticity and individual level heterogeneity can be judged, allowing examination of situations where estimation of mean quantities through Monte Carlo simulation (*e.g.*, running MBITES repeatedly) deviates from the deterministic approximation. It is not the goal of MBDETES to produce a full deterministic approximation of the complete spatial dynamics of MBITES, but for model checking and investigating the importance of stochasticity on the individual level.

The variables in MBDETES equations represent the density of mosquitoes that are in each behavioral state, and the parameters describe state transition probabilities and associated waiting times. Let a variable name denote the population density of mosquitoes in each state, where the variable names in MBDETES matches the code letter for the behavioral states and the postprandial resting period in MBITES: blood feeding (B), egg laying (O), searching for blood (F), searching for habitats (L), and post-prandial resting (R), for each site. Let $P_{XY}$ denote the proportion of mosquitoes transitioning from state $X$ to $Y$ after one bout, $1/\gamma_X$ denote the duration of time to complete one bout in state $X$, and let $\Lambda_F(t)$ and $\Lambda_B(t)$ denote the rates of mosquitoes emerging end entering into F and B states, respectively:

$$\frac{dF}{dt} = \Lambda_F(t) + \gamma_B P_{BF} B + \gamma_O P_{OF} O + \gamma_R P_{RF} R - \gamma_F (1 - P_{FF}) F$$

$$\frac{dB}{dt} = \Lambda_B(t) + \gamma_F P_{FB} F + \gamma_R P_{RB} R + \gamma_O P_{OB} O - \gamma_B (1 - P_{BB}) B$$

$$\frac{dR}{dt} = \gamma_B P_{BR} B - \gamma_R R \qquad (1)$$

$$\frac{dL}{dt} = \gamma_R P_{RL} R + \gamma_O P_{OL} O - \gamma_L (1 - P_{LL}) L$$

$$\frac{dO}{dt} = \gamma_R P_{RO} R + \gamma_L P_{LO} L - \gamma_O (1 - P_{OO}) O$$

Note that there are some more general cases of MBITES (*i.e.*, with mating and sugar feeding) in which behavioral state transitions can be formulated as a continuous-time Markov process and could thus be described by similar systems of equations. MBITES models that build up state memory over time in a mosquito, breaking the memoryless assumption of Markovian

systems (by depending on age, oogenesis, egg-batch size, or other variables which depend on previous states), could possibly be modeled, but they would require more complex systems of equations. In particular, the model of oogenesis in which re-feeding depends on previous blood meals would require modifying the state space to track egg batch size, or use integro-differential equations. We note also that it is possible to develop MBDETES models that are "spatial." While such models can be built, they are beyond the scope of this paper.

## Results

Behavioral state models, such as MBDETES and models developed in MBITES, are based on a detailed description of mosquito behaviors, behavioral states, and behavioral state transitions. These models do not supply the standard bionomic parameters. Instead, they show how mosquito bionomic parameters most relevant for pathogen transmission arise from the simple algorithms that drive the mosquito behavior. In addition, MBITES has additional built-in flexibility to show how these parameters are affected by geography (distribution of resources), ecology (interactions of mosquitoes with other biotic and abiotic elements of their environment), climate, and other factors external to the individual mosquito. The small set of summary statistics that has been traditionally used to describe aspects of mosquitoes relevant for transmission (Table 1) are thus emergent features of a complex interaction of mosquito behavior in an environment. In MBITES, these parameters arise naturally from mosquito behavior for a given place, shedding light into what aspects of ecology affect these population level summary metrics the most.

Here, using some models that were developed within the MBITES framework, we show how to compute these parameters and illustrate some basic features of mosquito ecology relevant for transmission. We also use MBDETES to verify MBITES (and *vice versa*).

### Vectorial capacity and bionomic parameters

VC in MBITES models can be computed in two ways. First, VC could be estimated by taking the product of the average bionomic parameters, using standard formulas (Table 1). Second, the bionomic parameters are not specified as parameters in MBITES, but they can be computed as a summary description of mosquito behavior through Monte Carlo simulation.

It has been shown that the traditional formula for VC can be reduced to just three terms—the number of emerging adult female mosquitoes, per human, per day ($\lambda$); the probability of mosquito survival through the extrinsic incubation period (EIP) of the pathogen ($P$); and the stability index ($S$) [11]. The Ross-Macdonald formula for the VC is equivalent to: $V = \lambda S^2 P$. The stability index appears twice because pathogen transmission requires a mosquito to take two distinct human blood meals—one to infect the mosquito and another to infect the pathogen's human host (after surviving the EIP and becoming infectious). Unlike the entomological inoculation rate, the VC does not rely on any information about the parasite reservoir in humans; that is, VC measures only the entomological capacity of a particular setting to sustain pathogen transmission, and is independent of prevalence of human infection [11].

**Expectations in MBITES and MBDETES.** One method to compute the bionomic parameters is by computing the expected transition probabilities from each state to all other states accessible from it. This discrete distribution can be calculated by following the sequence of Bernoulli events in Figs 2 and 3, for example, taking expectation values at each step. By averaging in this way, the $P_{X,Y}$ quantities for MBDETES can be computed from MBITES. The exact multinomial probability distribution over outcomes ($P_{X,Y}$) for each bout can also be calculated numerically without simulation, by summing the relative probability of each path through the bout; certain branch points are based on random sampling, such as blood meal size, in these

cases, we calculate the expectation with respect to the random variable by numerical integration (these functions can be found in the file https://github.com/dd-harp/MBITES/blob/v1.0/MBITES/R/MBDETES-Calibration.R in the MBITES package).

Using these methods, we can then compute quantities in MBITES or MBDETES linking estimates back to the bionomics commonly used as input (or derived as equilibrium) in Ross-Macdonald style models. From the single-bout transition probabilities and waiting times for each bout type, $P_{XY}$ and $T_X$, we derived formulas for the proportion surviving and the waiting time for surviving mosquitoes to make the transition from: $F$ to $B$, including loops back into $F$; from $B$ to $R$ including loops back into $F$ and $B$; from $L$ to $O$; including loops back into $L$; and from $O$ to $L$, including loops back into $L$ and $O$. The waiting time in each state $T_X$ is determined by parameter, and the inverse is the rate parameter $\gamma_X$ used for simulation in MBDETES (Table 2). These closed-form solutions for the means of these bionomics allow us to compare MBDETES to classical Ross-Macdonald parameters.

The inverse of this resting period to resting period waiting time maps onto the feeding rate parameter $f$ in the Ross-Macdonald model (see Table 1); $T_{R,R} \mapsto 1/f$. The rest to rest survival probability is also needed to link MBDETES to Ross-Macdonald parameters; this maps onto the probability of surviving through one feeding cycle, $P_{R,R} \mapsto e^{-g/f}$, or $\frac{-\ln P_{R,R}}{T_{R,R}} \mapsto g$.

**Monte Carlo simulation in MBITES.** In these simulations, there is a more direct way of computing the VC. For each mosquito from emergence to death, MBITES logs each activity bout, including the time, location, behavioral state, the values of other variables, the identity of every human host probed, and the identity of every host who gave a blood meal. Mosquito survival can be computed simply from the distribution of the mosquito age at death. Similarly, the overall feeding frequency can be computed from the distribution of time intervals between successive blood meals.

The formulas for VC arise from an anthropocentric concept that counts events happening to a human on a day. Here, we compute VC directly by following every bite occurring on a single person on a single day, and then summing all secondary bites by the initial biting mosquito separated from the initial bite by at least EIP days. To put it another way, the average VC is the number of pairs of human bites given by one mosquito that are separated by at least EIP days, summed over all mosquitoes and divided by the number of humans, reported per day. The distribution of VC uses the infecting human as the reference (thus VC is in units of human$^{-1}$ day$^{-1}$).

To compute the VC in this way, MBITES simulation output was summarized as follows: 1) the first bite in the pair must have been a blood meal; only blood feeding can infect a mosquito; and 2) the second bite in the pair included all events in which a human was probed (as parasites or pathogens usually enter the wound in a matrix of salivary proteins during probing); 3) the time interval between the two events was greater than or equal to the EIP; 4) the number of secondary bites is tallied over all mosquitoes by adding them to the human who was bitten on the first encounter; 5) the total was divided by the number of days. Because all events in the simulation occur at a set of sites in space, the spatial dispersion of VC can be calculated by simply attaching the distance between these secondary bites from the primary bite to each pair of bites. This level of realism is possible because probing and blood feeding are accounted for separately and accurately in MBITES.

Notably, because MBITES simulates the blood meal as a process of probing followed by blood feeding, some mosquitoes may be killed or interrupted after probing but prior to blood feeding, so there could be small differences in the computed quantities of VC from the classic Ross-Macdonald formula and computation via MBITES.

## Verifying MBITES and MBDETES

To address a challenge facing most complicated individual-based models, we have identified models in MBITES that are the stochastic analogues of models in MBDETES. The following sections illustrate how this was done.

**Bionomic parameters with MBDETES.** To compute some of the bionomic quantities in MBDETES, we set $F(0) = 1$ and $\Lambda_F(t) + \Lambda_B(t) = 0$ such that the equations track a cohort as it ages. Using these equations, we track the proportion surviving by age: $F + B + R + L + O$, the laying rate $\gamma_O(P_{OF} + P_{OB})O$, and the blood feeding rate, $\gamma_B P_{BR} B$. In practice, given these initial conditions, this set of equations can be numerically solved to derive bionomic parameters of interest.

Because the length of a feeding cycle is a crucial determinant of transmission potential, we developed another set of equations to compute the probability distribution of the time needed to complete a single feeding cycle. Our equations describe all histories taken by a mosquito just beginning a post-prandial rest ($R_1$) which could lead to one of two absorbing states: death or the next rest, which implies successful feeding ($R_2$). By numerically solving these equations, the distribution of feeding cycle lengths, conditional on survival can be computed. Details on the equations and computation are presented in Supplemental Information.

**Mapping MBITES onto MBDETES.** A challenge for most complex individual-based models is having a method for verification. Here, for a comparatively simple model developed in MBITES, there is a theoretical match to a system of ordinary differential equations formulated in MBDETES. The two models can be used for mutual verification.

We developed a model in MBITES that maps onto a set of equations in MBDETES under a specific set of conditions. First, we use a "trivial" landscape in which there were three haunts with symmetric movement probabilities: one haunt with only a blood feeding resource, one haunt with only an aquatic habitat, and one haunt with both types of resources (*i.e.*, a peridomestic haunt). Second, exponential distributions were used to sample all waiting times for behavioral state transitions. Third, re-feeding probability is only a function of blood meal size. We summarized the Monte Carlo simulation in MBITES by computing histograms, which were overlaid on top of density functions computed in MBDETES by numerically solving the system of Kolmogorov forward equations Eq (1).

The equations in MBDETES were parameterized following the method presented earlier (section Expectations in MBITES and MBDETES). Additionally, we set $\Lambda_F(t) = \Lambda_B(t) = 0$, and let $dD/dt = -dF/dt - dB/dt - dR/dt - dL/dt - dO/dt$, and initial conditions such that $F + B + R + L + O + D = 1$. Under these conditions the system of equations corresponds to the Kolmogorov forward equations for individual behavioral space trajectories, averaging over the three haunts. In this interpretation, the state variables $F, B, R, L, O, D$ represent the distribution of probability mass over the set of states a mosquito may belong to at any point in time $t$. If provided with an initial mass over states at $t = 0$, the numerical solution gives the time evolution of the probability to find a mosquito in any state at any time $t$.

In Fig 4 we show a comparison of MBDETES and MBITES for several bionomic parameters, where MBITES was simulated under a set of simplified conditions such that MBDETES correctly describes the predicted probability density functions of the parameters. Bionomic parameters for MBITES were computed from simulation output and compared to deterministic approximations from MBDETES by overlaying histograms over predicted density functions and comparing means (Fig 4). MBDETES was simulated by numerically solving the system of Kolmogorov forward equations Eq (1). Under this set of assumptions, the numerical results from MBDETES are matched by the results of the MBITES Monte Carlo simulation.

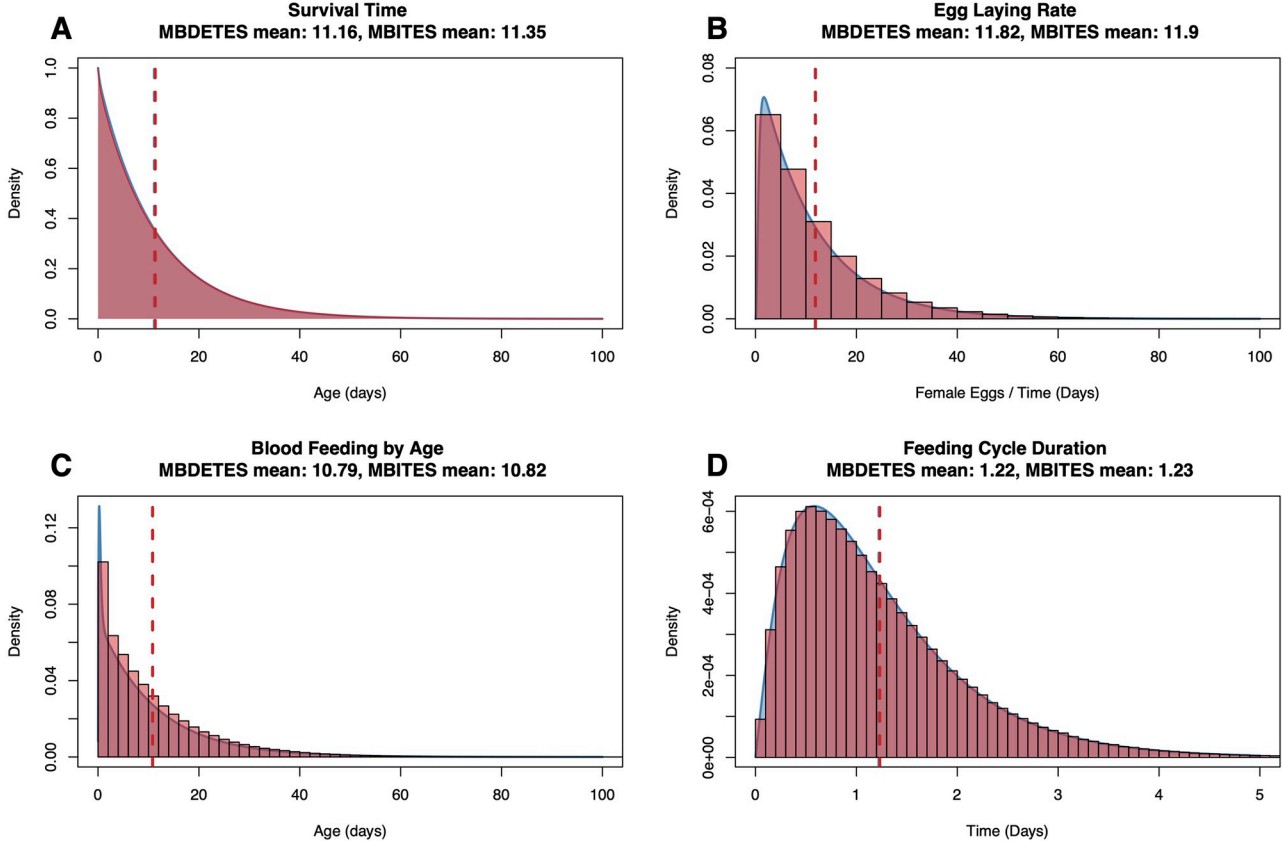

**Fig 4. Comparison of results from MBITES and MBDETES under restricted (Markovian) assumptions on waiting times and state transition probabilities.** B: Egg laying rate is the number of eggs laid, per female, per day. C: Blood feeding by age is the age distribution of mosquitoes taking bloodmeals. D: Feeding cycle duration is the time between post-prandial resting periods. In each panel, MBITES is summarized as a red histogram overlaid against the smooth density (in blue) predicted by MBDETES. All cases see excellent agreement, with MBITES fluctuating around MBDETES due to finite sample size of mosquitoes in the stochastic simulation.

## Dispersion in MBITES

Because the specific trajectory a mosquito takes is a random process that depends on the particular haunt containing the aquatic habitat from which it emerged, the complex interactions between a mosquito's internal behavioral state, the spatial arrangement of haunts and resources, and the movement of human (and non-human) blood hosts, it follows that the movement of mosquitoes on a landscape emerges from interactions among these components and algorithms. Averaging across all location-specific movement kernels gives a sense of how far mosquitoes are likely to travel (Fig 5), though the realized distribution for any particular simulation will differ depending on how often each route is used.

Dispersion in MBITES is a probability mass function describing the distances traveled as mosquitoes redistribute themselves among point sets: all distances traveled are drawn from the set of pairwise distances among sites. The dispersal kernels can be visualized directly for each site (*i.e.*, the probability mass on each distance), or from a simulation or overall: we compute the empirical cumulative distribution function (eCDF) of distances traveled; 2) fit a smooth curve to the eCDF; and 3) take the derivative of the smoothed eCDF. These smoothed kernels were estimated through the "lokern" package for R [26], and provide a rough estimate of how far mosquitoes will travel.

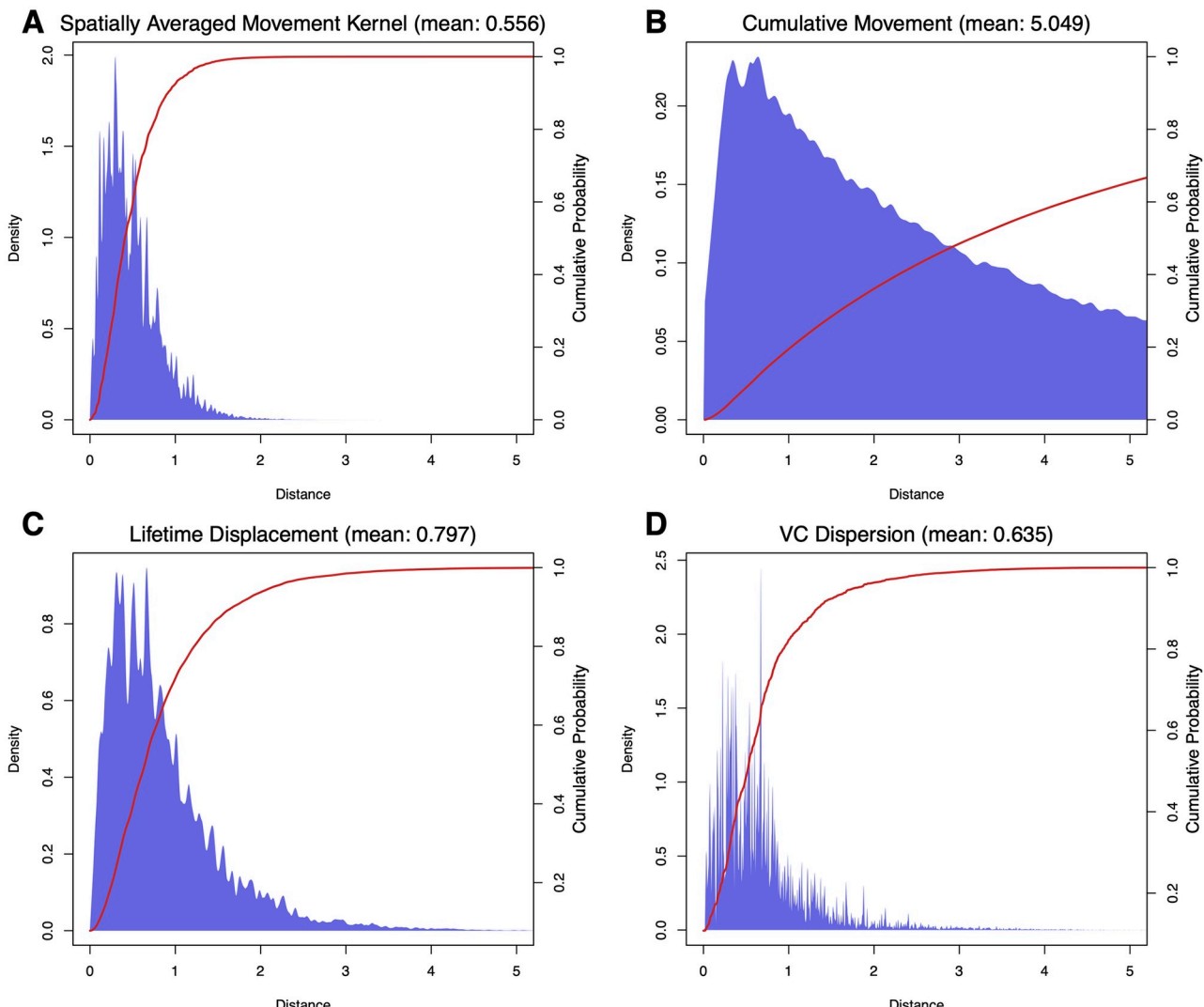

**Fig 5. Measures of mosquito dispersion.** Smoothed distribution (red line) and density (blue area) functions are displayed for summary statistics calculated for one particular landscape (50% peri-domestic habitats). A: The spatially averaged movement kernel is simply the probability of movement by distance, averaged over all haunts on the landscape. B: Cumulative movement, gives the distribution of total distance traveled by mosquitoes over their entire lifetime, and has a long right tail. C: Lifetime displacement is the absolute *displacement* of a mosquito, that is, the distance between the natal aquatic habitat they emerged from and the site at which their died. D: Dispersion of VC shows the distribution of secondary bites by distance, and follows closely absolute displacement of mosquitoes. All plots are calibrated to the same x-axis for comparison.

Distributions of mosquito movement by distance can also be directly calculated by Monte Carlo simulation through MBITES, allowing examination of characteristic scales of movement in the presence of specific models of human activity. Functionally relevant summaries of mosquito movement come from simulating MBITES and plotting these same kernels for spatial dispersion of VC as computed above (Fig 5). A key difference between these dispersal kernels and simulated empirical kernels is that the interactions between mosquitoes and humans will produce empirical kernels that are not necessarily the same as what one would obtain if we had simply used the dispersion kernels to smooth average bionomic parameters over space.

Detailed measures of how far mosquitoes disperse through space, transport parasites, or distribute eggs relative to their natal aquatic habitat can be calculated and described by

probability density functions. In Fig 5, densities for one particular landscape were calculated by taking the empirical cumulative distribution function (CDF) from simulation output, applying a smoothing algorithm to estimate a smooth CDF, and differentiating to estimate a probability density function (PDF) [26]. Four measures of dispersal were computed. First, the upper left density is the average site to site movement kernel, and shows the average "one hop" distance a mosquito will make during a activity bout if it leaves to search. Cumulative dispersion (upper right) is the average distance traveled by a mosquito summing over all hops in its lifetime. Absolute dispersion (lower left) summarizes the mean *displacement* of mosquitoes; that is, the distance between their natal aquatic habitat they emerge as adults from and the site of death. Most relevant for pathogen transmission is VC, for which the average distance between each pair of secondary bites is a measure of the capability of the local ecology and vector population to sustain pathogen transmission spatially. Dispersion of secondary bites tracks much more closely measures of absolute dispersion rather than cumulative distance traveled, which will be explored in greated detail below.

## Co-Distribution of resources

Searching behavior, in which mosquitoes take long range flights or many short hopping flights to look for resources, plays a crucial role in mosquito behavior but also contributes an important source of mortality that could be avoided if local resources are plentiful. The distribution of local resources structures mosquito movement, therefore, holding all else constant, the amount of time a mosquito spends searching for resources should affect the various summary bionomics that describe the mosquito population and its ability to be effective vectors of pathogens. To illustrate the effect of distribution of resources on the bionomic parameters, we conducted a set of experiments showing the influence of resource co-distribution on summary bionomics.

The set of *in-silico* experiments was designed to explore how bionomics changed as a function of the availability of peri-domestic habitats, which describes the proportion of blood feeding at haunts that have a viable aquatic habitat in the local vicinity; in terms of simulation, a peri-domestic haunt has both types of resources present. The proportion of haunts with viable aquatic habitats nearby may differ depending on local distribution of resources. The extent to which the haunts get visited depends the extent to which there is significant correlation between the set of points at which mosquitoes can locate suitable blood meal hosts and the set of points at which mosquitoes can oviposit egg batches.

**Peri-domestic simulation.** To examine the effect of peri-domestic habitats on generated mosquito bionomics, 26 resource landscapes were generated, each containing 250 blood feeding haunts, and 250 aquatic habitats. Peri-domestic habitats (the percent of haunts that contained both resources), ranged from 0% in landscape 1, to 100% in landscape 26 (Fig 6). Put another way, in landscape 1, each site contained *either* a blood feeding haunt *or* an aquatic habitat, whereas in landscape 26, each site contained *both* types of resources. In all landscapes the total number of resources was held constant as described above. Spatial variance in location of haunts was simulated by choosing 25 parent points for blood feeding haunts, and then scattering 9 offspring blood feeding haunts around each parent, for a total of 250 haunts in 25 clusters. In the simulation where peri-domestic habitats was held at 0%, aquatic habitats were simulated independently using the same algorithm. For all other landscapes, overlap was simulated by selecting some fraction of the total number of aquatic habitats and attaching them randomly to blood feeding haunts.

In each simulation, all biological parameters were held constant such that the only varying parameter was the spatial arrangement of resources. In addition, mosquito emergence rates

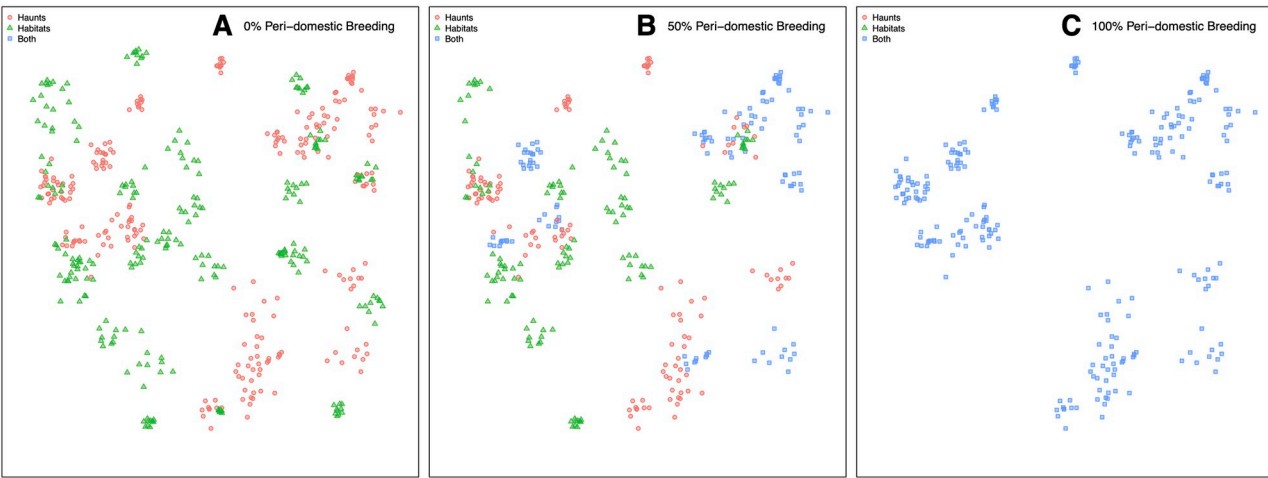

**Fig 6. Simulated landscapes.** 3 simulated landscapes at A: 0%, B: 50%, and C: 100% peri-domestic habitats. Haunts that contain only blood feeding haunts are plotted as red circles, haunts that contain only aquatic habitats are plotted as green triangles, and those haunts that contain both types of resources are shown as blue squares (*i.e.*, peri-domestic habitats). Dispersal kernels were calibrated as if this was an area of about 100 km$^2$.

from each aquatic habitat was also held constant, such that on average, each habitat produced one adult female per day. We acknowledge that in the absence of population dynamic feedback effects the model may not correspond to any true ecological system. For the specific analysis considered here, we are specifically interested in how spatial arrangement of resources affects how mosquitoes distribute their time across behavioral states, and the effect it has on commonly used bionomic parameters. We intend to revisit the question of population dynamic feedback effects in future research.

**Sensitivity of parameters to peri-domestic habitats.** To study sensitivity of bionomics to peri-domestic habitats, we calculated all bionomic parameters for each of the 26 simulated landscapes. Of particular interest was the change in VC as a function of peri-domestic habitats. As the percentage of blood feeding haunts with associated peri-domestic mosquito aquatic habitats increased, mean VC (measured in units person$^{-1}$ day$^{-1}$) increased by 2 orders of magnitude (Fig 7). While the stability index also increased as a function of peri-domestic habitats, the absolute difference between *S* evaluated at 0% and 100% peri-domestic was smaller, only increasing by one order of magnitude (Fig 7). Importantly, we currently assume no competition among mosquitoes during oviposition, but we note that nonlinear competition has been observed and can easily be included in the model, at the cost of complicating analytic analysis [23].

Significant differences were also observed in how mosquitoes transitioned between and partitioned their time over the set of behavioral states. To quantify, we computed empirical state transition matrices for each landscape, consisting of the behavioral states *F, B, R, L, O*, plus an absorbing state *D* for death. For each mosquito, all jumps between states were tabulated in the transitions matrix **T**, which was then normalized such that **T** *e*′ = 1. These empirically estimated Markov transition matrices are displayed as chord diagrams in Fig 8 for 0%, 50%, and 100% peri-domestic habitats. Transitions between states are represented by colored edges, where width of the edges is proportional to the probability of that transition. Colored areas on the circumference of the diagram represent the states, where size is proportional to that element of the quasi-stationary distribution for that landscape (the mean proportion of time spent in that state, conditional on survival). From 0% to 100% peri-domestic habitats the

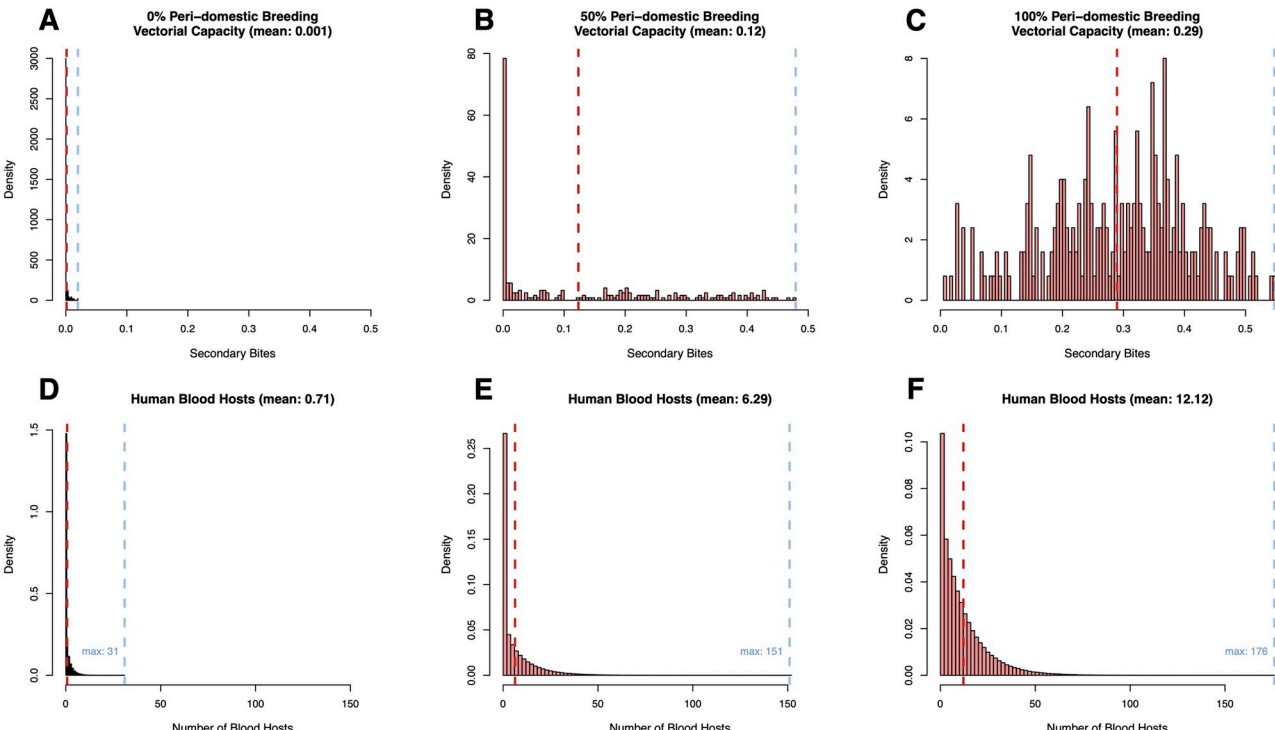

**Fig 7. Vectorial capacity.** In MBITES, vectorial capacity (VC) is computed directly as the average number of infectious bites (*i.e.*, probing) arising from all the mosquitoes blood feeding on a single human on a single day; it is effectively the number of pairs of events where a blood meal by a mosquito is followed at least EIP days later by that same mosquito probing in attempt to feed on a human, measured per human, per day. Summary VC A,B,C; and number of human blood meals per mosquito over its lifespan (D,E,F; referred to as the *stability index* by Macdonald) are shown by column for 0%, 50%, and 100% peri-domestic habitats. Each histogram gives the distribution of VC or the number of human blood hosts across mosquitoes for that percent peri-domestic habitats.

probability of a blood feeding search leading to another search dropped from 0.90 to 0.17 in the most resource-rich setting, which we consider as easy access to local resources. To determine how mosquitoes partitioned their time across these states, for each normalized matrix **T**, we calculated the quasi-stationary distribution across the transient states *F*, *B*, *R*, *L*, *O* [27]. This distribution describes how a mosquito spends its time, conditional on survival. At 0% peri-domestic, mosquitoes spend near 77% of their time prior to absorption searching for blood meals, and close to 12% of their time searching for suitable aquatic habitats to oviposit (Fig 8D). At 100% peri-domestic, these proportions drop to a mere 11% and 6%, respectively (Fig 8F).

For each of the 26 resources landscapes, we also calculated mosquito lifespan, stability index, duration of feeding cycle, blood feeding rate, absolute and cumulative mosquito dispersion, VC, and spatial dispersion of VC (Figs 9 and 10). As the resources were rearranged to increase peri-domestic habitats, lifespan, number of blood hosts, and blood feeding rate increased, while duration of the feeding cycle decreased. Certain bionomics, such as blood feeding rate, and length of feeding cycle tended to reach a plateau after peri-domestic habitats increased beyond about 20%, as the mosquito life cycle does not allow these values to change indefinitely. Cumulative mosquito movement (the cumulative distance traveled in all activity bouts) decreases as peri-domestic habitats increases; as the environment becomes more resource rich, mosquitoes do not need to travel as far to fulfill their biological intent (their current behavioral state). In contrast, absolute dispersion (the *displacement* of a mosquito between

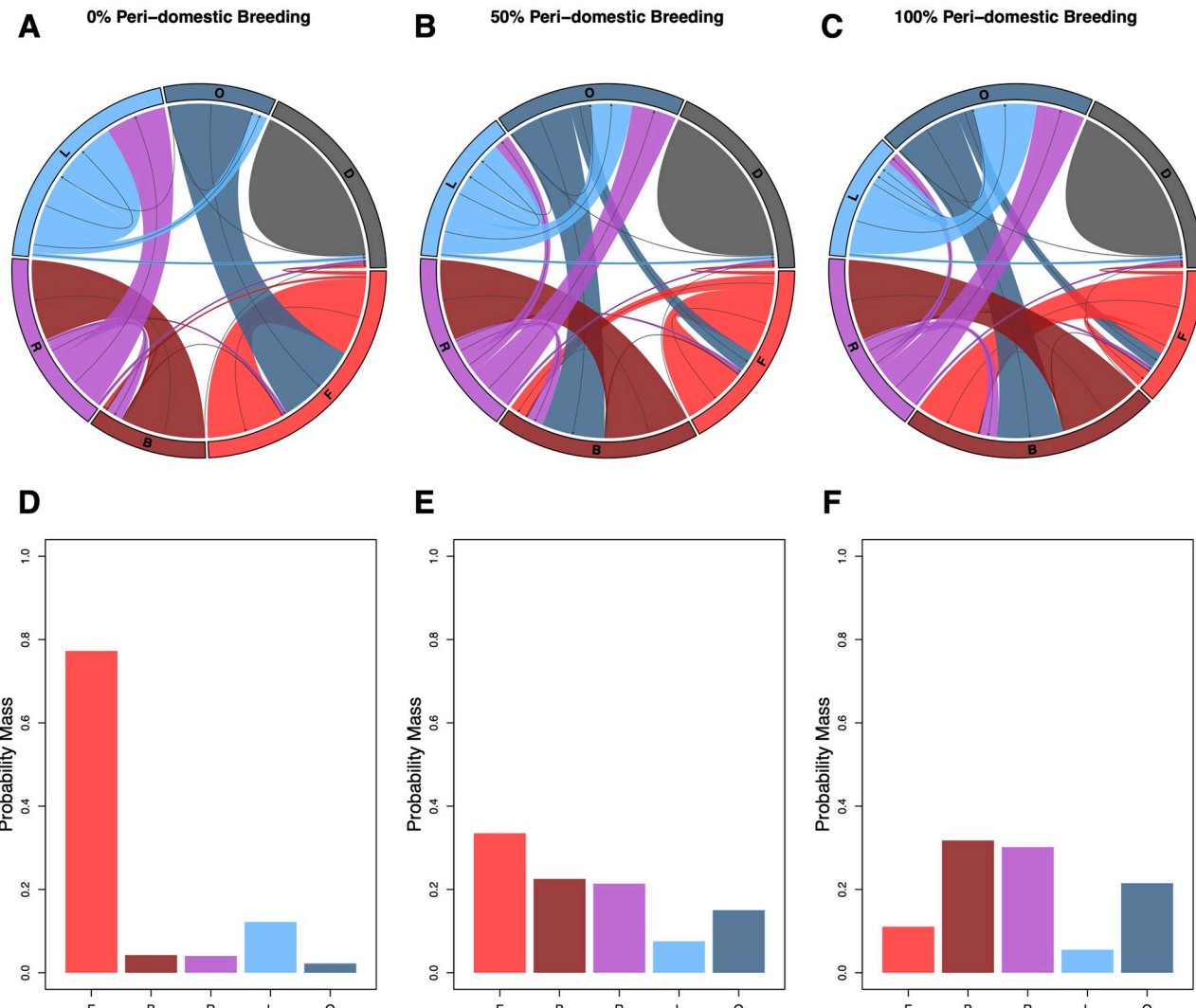

**Fig 8. Behavioral state distribution.** Chord diagrams showing the empirical state transition matrices for three of the 26 experiments: A: 0%, B: 50%, and C: 100% peri-domestic habitats. These were calculated for each experiment by summing transitions for each mosquito between two states and then averaging to produce a Markov transition matrix. The width of the directed edges between each behavioral state is proportional to the probability of that transition, and the area on the perimeter of the circle labeled for each state is proportional to the mean time spent in that state. The three chord diagrams are accompanied below (D-F) by quasi-stationary probability distributions which give the asymptotic distribution of how a mosquito spends time across behavioral states conditional on survival.

the natal habitat they emerged from and where they died) stays relatively constant as a function of peri-domestic habitats. This apparent discrepancy between the simulation results and intuition can be understood by noting that while the percent of haunts that were considered peri-domestic changed between the 26 landscapes, the spatial arrangement of haunts did not (distances and clusters were preserved). Holding these spatial characteristics constant implies that observed differences are due solely to changes in search patterns of mosquitoes as resources become more or less locally dense.

As noted in Fig 7, VC increased dramatically as peri-domestic habitats increased. It is notable that at very low levels of peri-domestic habitats, the mean is strongly affected by a few outliers generating large numbers of secondary bites; the effect of these outliers is dampened as

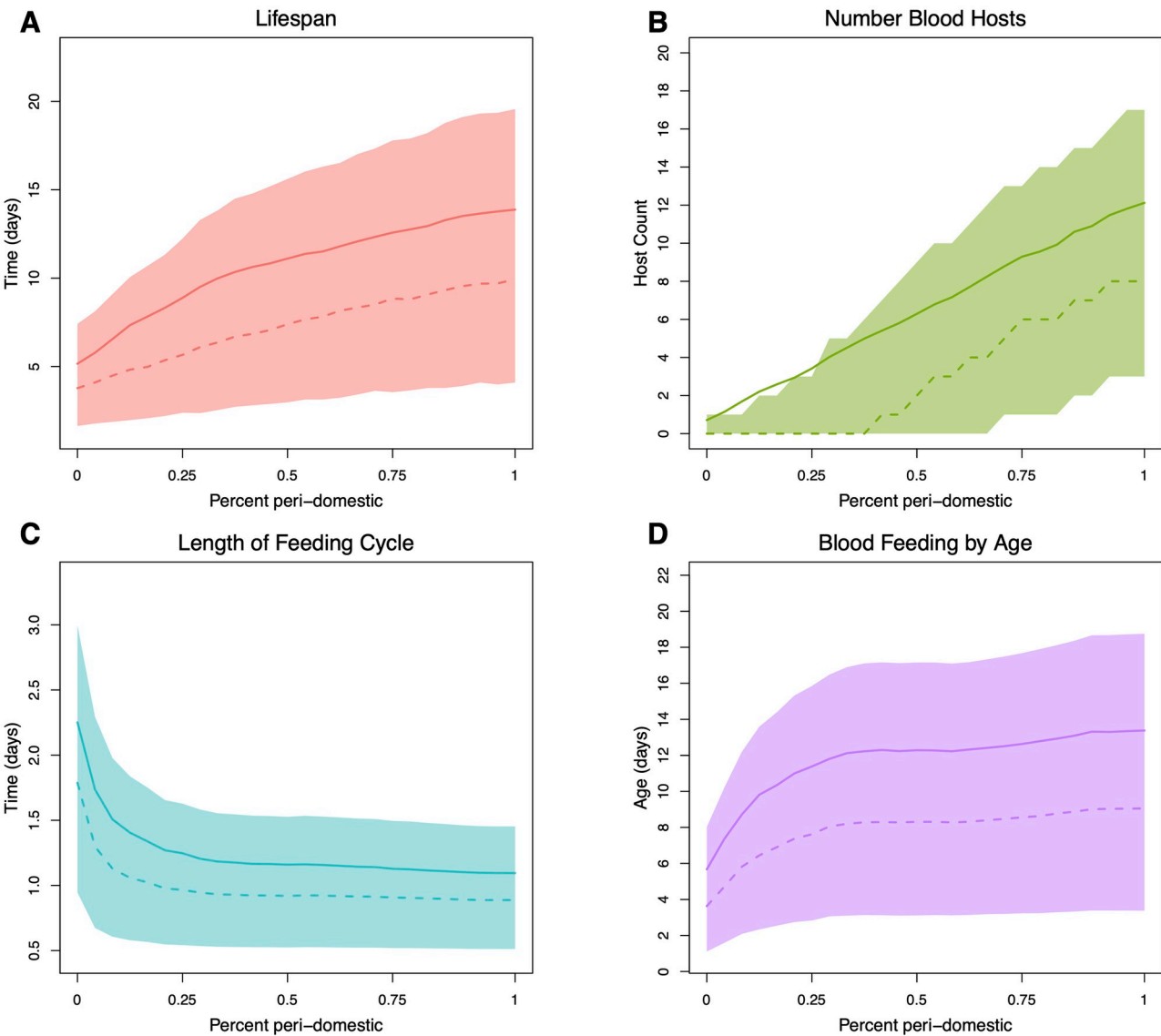

**Fig 9. Binomic parameters.** Simulations in MBITES illustrate that all of the bionomic parameters are sensitive to the proportion of peri-domestic habitats, which gives a measure of how frequently a mosquito must search. The x-axis of each plot ranges from 0% to 100%, and each summary bionomic parameter is plotted as mean (solid line), median (dashed line), and the shaded area covers the 20-80% quantile range of the data. The distribution of number of blood hosts B: exhibits significant right skew, such that the mean exceeds the 80% quantile at low proportion peri-domestic breeding habitats. Because the simulations are stochastic, the exact number of mosquitoes from which Monte Carlo estimates of the bionomic parameters were computed varied somewhat over the 26 landscapes, the mean was 456,579 mosquitoes with a standard deviation of 754 mosquitoes.

peri-domestic habitats increases and more humans contribute to secondary biting (Fig 10). Spatial dispersion of secondary bites remains relatively constant across landscapes; and follows closely the absolute dispersion of mosquitoes. This is primarily due to two effects. First, because VC was calculated assuming a pathogen with an EIP of 10 days, only mosquitoes that survived at least that long would be able to contribute secondary bites, so the site that secondary bites would be successfully delivered to would be close to the site of the eventual death of the mosquito. Additionally, because the spatial characteristics of the landscape were not affected to a large extent by the rearrangement of resource overlap, although the absolute value of VC changed dramatically, its spatial dispersion did not.

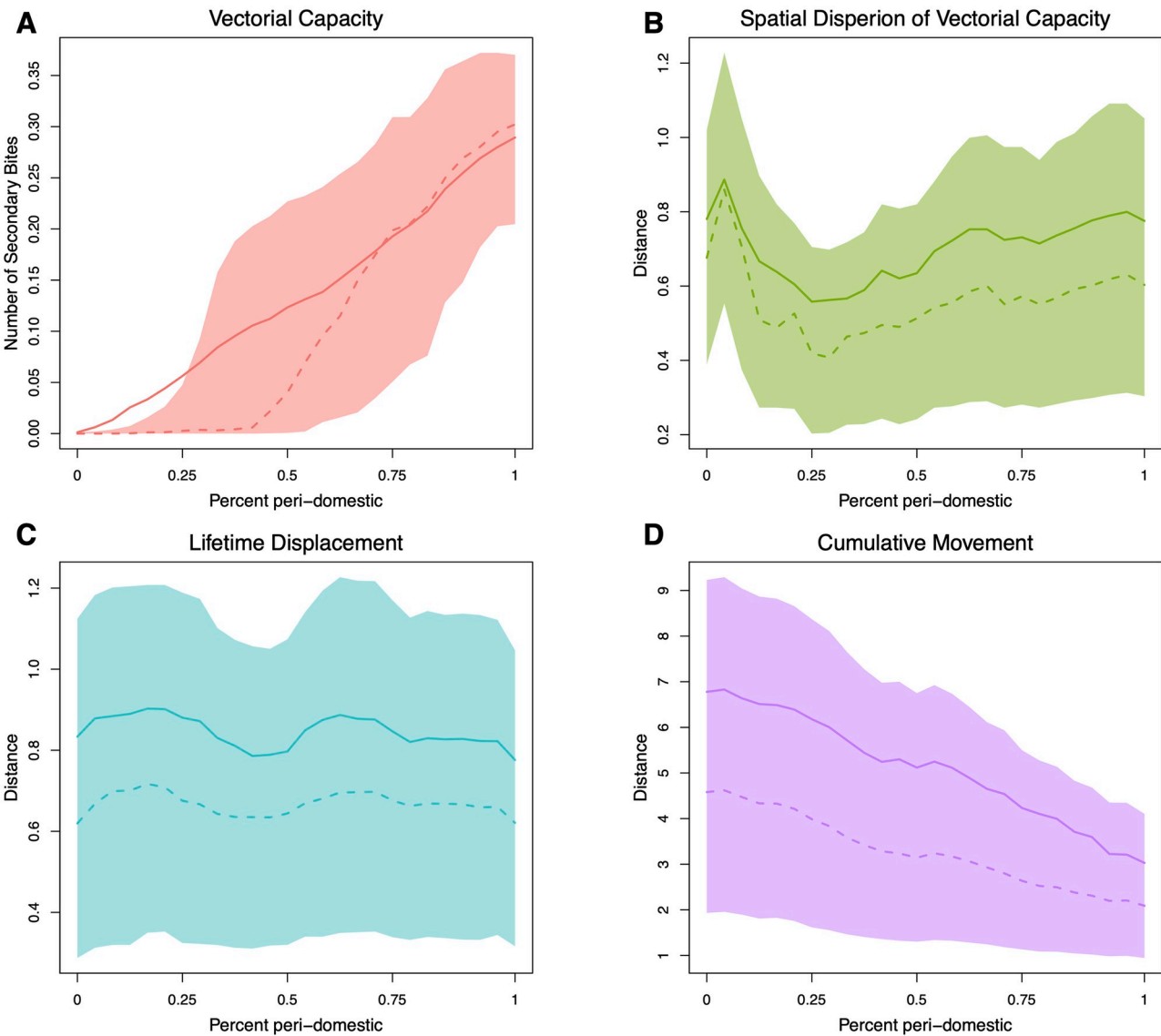

**Fig 10. Dispersion and movement parameters.** In MBITES, vectorial capacity (VC) and its dispersion are highly sensitive to the proportion of peri-domestic habitats. Interpretation of axes follows Fig 9, and each summary bionomic parameter is plotted as mean (solid line), median (dashed line), and the shaded area covers the 20-80% quantile range of the data. A: Number of secondary bites produced increases dramatically as a function of peri-domestic habitats. B: Spatial dispersion shows no strong trend however, due to the strong clustering of haunts in the landscape (it largely follows the trend of absolute lifetime displacement (C), as opposed to cumulative movement (D). At low percent peri-domestic breeding habitats, significant right skew in the distribution of VC pulls the mean above the 80% quantile.

## Discussion

The sensitivity of standard bionomic parameters, including lifespan, stability index, and vectorial capacity, to the proportion of habitats that are peri-domestic has implications for disease transmission potential as well as response to control. By rearranging the spatial arrangement of the same resources, we showed that increasing proportion of peri-domestic habitats causes proportionate increase in VC and affects the dispersion of these potential secondary bites. Each parameter that strongly affects VC is sensitive to the proportion of peri-domestic breeding sites (Fig 9, lifespan, mean age of blood-feeding mosquitoes, and length of gonotrophic

cycle). This was alluded to in an earlier mathematical model [28] but this is the first demonstration of sensitivity to peri-domestic breeding in a stochastic agent-based model. The strong dependence of bionomics upon the proximity of breeding sites to blood feeding haunts has encouraging implications for larval source management based vector control. In the case where enough such habitats are findable, sustained larval source management could potentially reduce the proportion of peri-domestic habitats and consequently reduce VC. Adult vector control methods could also be preferentially deployed based on the proportion of peri-domestic breeding sites. The distribution of a mosquito's lifespan across various behavioral states changes substantially as a function of this proportion (Fig 8). One implication is that, if mosquitoes were sugar feeding more frequently than usual in a resource sparse environment where behavior is dominated by long searching flights (as in 0% peri-domestic breeding example), attractive toxic sugar baits (ATSB) could potentially have a large impact. Alternatively in settings where more time is spent in and around human dwellings, traditional tools such as long-lasting insecticide treated nets (LLINs) and indoor residual spraying (IRS) may remain the best choice for vector control.

MBITES was developed as a framework for building models of sufficient complexity to examine whether the current widely used small set of entomological parameters is adequate to the task of identifying critical features of transmission and informing control. Despite the model's enormous complexity, what became obvious was that much of what was occuring during a single activity bout can be summarized by simple outcomes and their accompanying behavioral state transitions: death, failure or success to feed or oviposit, and the decision to stay or search. MBDETES is our method to grapple with the complexity of MBITES mathematically. While MBITES is a stochastic agent-based model with upwards of 40 parameters, MBDETES has 18 parameters and is expressed as a set of differential equations. One result from this rigorous mapping of detailed behavioral algorithms onto aggregate probabilities in MBDETES, was that much of the individual-level stochasticity in a bout was only important insofar as it affected aggregate transitions, though it is possible more complex models would show greater effect. Landscape details and mosquito responses to these details, on the other hand, proved to be enormously consequential for simulated outcomes. If one seeks to further reduce complexity, the result is the Ross-Macdonald model with five bionomic parameters, and the derived quantity of vectorial capacity summarizes transmission potential in a single expression.

A relevant lesson emerging from our analysis is how difficult it would have been to try and piece together estimates of mosquito bionomic parameters by measuring just directly observable aspects of mosquito behavior. The large number of parameters describing the outcome of a bout suggests the relevance of any particular direct observation of mosquitoes is only meaningful when it is measured along with all other parameters, and the importance of any particular behavior for transmission would thus probably differ by context. Our simulations have shown that it is important to challenge conventions and refine models. Estimated vectorial capacity is likely relevent for transmission only if measures of dispersion of infectious bites are taken into account [29, 30].

Despite the compelling logic and parsimony of the Ross-Macdonald model, do models of this type really capture all of the essential features of a time and place? Even if parsimony were the only measure of quality, there must be some way to determine the appropriate level of model complexity, which implies the necessity of building at least some non-parsimonious models for comparison. The functionality that is potentiated by MBITES and MBDETES comes with a cost; their parametric complexity makes these models suboptimal for tasks that demand parsimony. As with any complex, mimetic simulation model, issues of over-parameterization and limits of available field data to provide suitable parameter values arises. One way

to parameterize such models is by assuming some specific behavior has evolved optimally to some ecological context, and then back-solving for parameters which would lead to such behavior [31]. Another is *calibration*, where the model is fit to data by either statistical or more *ad hoc* approaches. However, another purpose of complex models is *sensitivity analysis* (SA). For a highly detailed model that reflects, as best as possible, existing entomological and ecological knowledge, SA methods may be employed in a variety of ways to identify non-obvious ways that mosquito populations could vary, and to help guide future empirical data collection which could best reduce uncertainty in some aspect of model output. Such knowledge can be quite relevant for planning intervention and control in natural systems, in addition to more basic scientific interests [32, 33]. SA is most powerful when used to evaluate how parameters, alone or in concert, produce qualitatively different model responses. Given a response of interest, Monte Carlo methods can help detect parameter interaction and the differing scales at which parameters wax or wane in importance, which can reveal unusual model behavior, identify possible control strategies, and improve interpretation of how models respond to perturbation [34, 35]. While extensive Monte Carlo simulations to perform SA may seem daunting, there is much active work in the field, including statistical emulation, efficient computer-based experimental design, and advanced history matching techniques [36–38].

Using individual-based models developed in MBITES, we have shown how bionomic parameters, vectorial capacity, and dispersion of mosquitoes and their bites arise from a set of mosquito behavioral algorithms in response to the distribution of resources and their spatial arrangement. Standard models lack the ability to comprehensively explore these questions, and while existing models offer compelling evidence of spatial effects [39], MBITES offers a higher level of individual and spatial resolution. In these models, search and dispersion are strongly affected by the co-distribution of resources [40], and the contextual factors affecting the frequency of search strongly affect vectorial capacity. The importance of ecological context was expressed eighty years ago by Hackett [41]:

> *Everything about malaria is so moulded by local conditions that it becomes a thousand epidemiological puzzles. Like chess, it is played with a few pieces but it is capable of an infinite variety of situations.*

This has been quoted frequently by malariologists, but it has been difficult to reconcile Hackett's view picture with the elegant concepts of VC and entomological inoculation rates, which concisely summarize the factors that are most likely to affect transmission. VC has a virtual hegemony in mechanistic models (where EIR is a derived concept), due partly to its simplicity and sound logic [13]. If the challenges of malaria transmission dynamics and control are best understood as a collection of puzzles to be solved and not effectively summarized by VC or EIR, then what other aspects of vector biology matter? The view of malaria as a chess game has, perhaps, been most apparent in the variable responses of vector control, where small behavioral differences among species have affected the rates of contact with interventions and thus the outcome of control. Our results suggest that fine-grained heterogeneity in movement driven by the co-distribution of resources shape mosquito dispersion, and that among-individual differences in VC and its associated dispersion will also have strong affects on transmission.

In retrospect, it is remarkable that investigation of pathogen transmission by mosquitoes has been so stable since it was jump-started by an intuitive leap by Ronald Ross [42]. Entomological work that followed over the next decades iteratively refined Ross's ideas culminating in a fully fledged theory of transmission [4, 6], including field methods to measure a handful of

relevant parameters. After more than a century of studying mosquito behavior, medical entomologists have assigned approximate bionomic parameter values to most of the parameters for most of the dominant vector species that transmit human infectious diseases. These have been assembled from hundreds of studies conducted in various ways over several decades. Biology and genetics constrain the behaviors giving rise to some of the differences in the average value of parameters assigned to some species. Our simulations show these bionomic parameters must also be partly determined by local resources and ecology. Consistent with Ronald Ross's original conception of *a priori* models as methods to assimilate disparate data, models like MBITES can help synthesize conclusions about transmission and control from novel sources of data, including entomological, genetic, ecological, and spatial. Despite this, existing studies do not adequately characterize a vector species across multiple settings. Studies are generally too heterogeneous in their design and implementation to be comparable across settings [43]. Few studies have identified systematic differences in mosquito bionomics looking across ecological contexts, perhaps because there has been no theory to suggest what they should look for. Neither evidence nor theory alone could provide a sufficient basis for making an educated guess about how the values of bionomic parameters vary in a different ecological setting or the effect modification of setting on vector control. These highly mimetic models of mosquito behavior can be used to set priorities in the study of pathogen transmission by mosquitoes, help shed light on the effect sizes of vector control, and explain heterogeneity in the outcome of control studies. In providing a framework for re-examining mosquito behavior and perhaps forging a new synthesis of ecology and behavior, these behavioral state models provide an *in silico* laboratory to fill some of the gaps required to understand and synthesize much of the data on mosquitoes that is not directly related to estimation of the basic bionomic parameters.

The mathematical and computational framework for simulation and analysis that is presented here can be used to investigate a broad range of questions about the interface between mosquito biology and life-history traits, the local ecology, and vector control. This framework was designed to evaluate heterogeneity and complexity through simulation, rather than through *ad hoc* approximation. While this approach has some disadvantages over parametrically simple models, it fills an important need by providing a way of testing whether those simple models cover all the relevant phenomena. MBITES makes it possible to systematically investigate whether the behavior of individual-based models ever deviates from the behavior of parametrically simpler models of transmission, such as the Ross-Macdonald model or MBDETES. The high degree of realism can also provide other functionality, such as power calculations for randomized control trials, investigation of bias and accuracy of field methods, *in silico* investigation of the interactions among vector-based interventions, and calibration of effect sizes versus coverage for vector-based interventions.

## Supporting information

**S1 Text. The S1 text (MBITES supplement) provides a complete description and figure of the modified MBDETES model used to compute the probability distribution of feeding cycle length presented in the section bionomic parameters with MBDETES.**
(PDF)

## Acknowledgments

We wish to thank Roly Gosling and Allison Tatarsky for their valuable input.

## Author Contributions

**Conceptualization:** Sean L. Wu, Héctor M. Sánchez C., John M. Henry, Daniel T. Citron, Qian Zhang, Biyonka Liang, Amit Verma, Derek A. T. Cummings, Arnaud Le Menach, Thomas W. Scott, Anne L. Wilson, Steven W. Lindsay, Catherine L. Moyes, Penny A. Hancock, Tanya L. Russell, Thomas R. Burkot, John M. Marshall, Samson Kiware, Robert C. Reiner, Jr, David L. Smith.

**Formal analysis:** Sean L. Wu, Héctor M. Sánchez C., John M. Henry, Daniel T. Citron, Biyonka Liang, David L. Smith.

**Investigation:** Sean L. Wu, John M. Henry, Daniel T. Citron, Arnaud Le Menach, Penny A. Hancock, David L. Smith.

**Methodology:** Sean L. Wu, Héctor M. Sánchez C., John M. Henry, Daniel T. Citron, Biyonka Liang, Amit Verma, Derek A. T. Cummings, Arnaud Le Menach, Robert C. Reiner, Jr, David L. Smith.

**Project administration:** Kelly Compton.

**Resources:** David L. Smith.

**Software:** Sean L. Wu, Héctor M. Sánchez C., Qian Zhang, Amit Verma, Derek A. T. Cummings, David L. Smith.

**Supervision:** Kelly Compton, David L. Smith.

**Validation:** Sean L. Wu, Thomas W. Scott, Anne L. Wilson, Catherine L. Moyes, Tanya L. Russell, Thomas R. Burkot, John M. Marshall, Robert C. Reiner, Jr, David L. Smith.

**Visualization:** Qian Zhang, David L. Smith.

**Writing – original draft:** Sean L. Wu, Héctor M. Sánchez C., David L. Smith.

**Writing – review & editing:** Sean L. Wu, Héctor M. Sánchez C., John M. Henry, Daniel T. Citron, Kelly Compton, Biyonka Liang, Amit Verma, Derek A. T. Cummings, Arnaud Le Menach, Thomas W. Scott, Anne L. Wilson, Catherine L. Moyes, Penny A. Hancock, Tanya L. Russell, Thomas R. Burkot, John M. Marshall, Robert C. Reiner, Jr, David L. Smith.

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
