## [Decision Letter · Decision Letter 0]

2 Jan 2020

Dear Dr Smith,

Thank you very much for submitting your manuscript, 'Vector bionomics and vectorial capacity as emergent properties of mosquito behaviors and ecology', to PLOS Computational Biology. As with all papers submitted to the journal, yours was fully evaluated by the PLOS Computational Biology editorial team, and in this case, by independent peer reviewers. The reviewers appreciated the attention to an important topic but identified some aspects of the manuscript that should be improved.

We would therefore like to ask you to modify the manuscript according to the review recommendations before we can consider your manuscript for acceptance. Your revisions should address the specific points made by each reviewer, particularly those raised by Reviewer 1 around the issues of parameter estimation and possible use fo the model for sensitivity analysis.  We note that the two reviewers differed somewhat on the readability of the manuscript, and Reviewer 1 suggested that it might be reduced in scope.  We defer this decision to you, but suggest that, if you opt to keep the current scope, you may wish to clarify the intended purpose of including details in the model that the field is perhaps not yet able to parameterize with confidence. 

- Supporting Information uploaded as separate files, titled 'Dataset', 'Figure', 'Table', 'Text', 'Protocol', 'Audio', or 'Video'.

We hope to receive your revised manuscript within the next 30 days. If you anticipate any delay in its return, we ask that you let us know the expected resubmission date by email at ploscompbiol@plos.org.

Sincerely,

James Lloyd-Smith

Associate Editor

PLOS Computational Biology

Virginia Pitzer

Deputy Editor

PLOS Computational Biology

[LINK]

Reviewer's Responses to Questions

**Comments to the Authors:**

Reviewer #1: The authors have described and put together a model that breaks down the mosquito gonotrophic cycle in great detail and places this in a spatial context of a landscape comprised of resources (blood hosts, larval development sites, and resting sites). They present both an individual-based model and a companion model based on ordinary differential equations and explore a number of questions, such as how the co-distribution of mosquito resources affects emergent properties such as vectorial capacity. The development of the model and the resulting manuscript are clearly the result of lots of careful thought and hard work. That the code itself is apparently being made publicly available is to be commended. I hope that consideration of the following points might help improve the manuscript further.

The manuscript itself is rather dense. While that’s fine, I think it’s worth considering if the authors aren’t trying to pack too much in. The manuscript provides a rather detailed overview of the model structure and the rationale behind all the various components. This includes quite a bit of discussion of additional types of states, resources, and hazards (sugar sources, mating, immature dynamics, local hazards, control methods, including ovitraps) or potential modifications (e.g., modifying the attractiveness of a larval development habitat if larvae are already present, infection with pathogens), that are already part of the model or may be added at some point in the future. A downside of providing all this information is that there are still lots of questions about the parameterization of the model that are not given as much attention as they could have (see questions below). The other part of the manuscript is an investigation of a model that considers host-seeking and feeding and oviposition, but not all the above-mentioned additional complexities. Limiting the description of the model itself to the actual components that are included in the model that is interrogated in this paper would make the whole more readable and perhaps free up some space to provide additional detail on parameterization (one could also easily imagine the model description itself, with a vignette aimed at potential users of such a model, being published as a separate paper).

As indicated, while wonderful that the authors are grappling with all this exquisite behavioral detail, it raises the question whether the parameterization of the model actually reflects our (lack of) understanding. For instance, if all hosts in a queue are given an attractiveness weight, what is this based on? Do we have studies that have compared the relative attractiveness of different humans as well as variation in attractiveness of different cows, and tested below which level of human attractiveness a mosquito will switch to a cow (or some other non-human animal)? Or whether there is a threshold level of attractiveness of a non-human above which a mosquito will consider biting that individual? It might well be a gap in my knowledge, but I don’t really think those experiments have been performed yet at the level you would need them to be to inform the parameter estimates.

Likewise, mosquitoes are described by a set of characteristics, including physical and physiological condition (e.g. wing tattering), energy reserves, blood meal size, and a set of variables related to sugar feeding and mating, among others. This is again admirable, but what are the parameters based on? For energetic reserves I imagine this was based on Briegel’s flight mill studies, but wing tattering and its cumulative effect on fitness – what reference is given there? (Perhaps this is mentioned in the appendix? It wasn’t available via Manuscript Central)

The behavioral transitions appear equally difficult to put reasonable values on – for instance, on p12 the choice whether or not to give up on a local patch and move elsewhere is said to occur when the resource is absent (makes sense) or after multiple failed attempts (how many, with what probability? What is this informed by) or, if the host is not suitably attractive the mosquito might leave as well. But do we really know enough about variation in attractiveness and whether a mosquito is likely to give up on a host in the absence of other hosts? I am reminded of another paper, by Ma & Roitberg (2008), that similarly breaks down mosquito behavior into the smallest amount of activity (and might be worth citing, actually) but which gets around the issue of placing estimates on these behavioral decisions by assuming optimal behavior and using backwards induction to solve for these.

While in the discussion you point to mosquito behavior having been studied for more than a century (L738), I would argue that the kind of questions posed here (to what extent is the outcome of a behavior dependent on or modified by environmental conditions) still haven’t been all that well-studied, instead rather focusing on a handful of relevant parameters (l780). My personal view is that while foraging ecology had a tremendous influence on biology and ecology overall, in mosquito research this field has been almost entirely neglected. On L809 it is stated that the current models provide an in silico lab to fill some of these gaps in understanding, but why instead not perform a standard sensitivity analysis with the intent to see which of these small behavioral details would be worth focusing on? To me, the most exciting part of this model would be to see what kind of behavioral ecology studies we should be engaged in to really validate this model and allow it to be useful for predictions regarding vector control.

Some additional minor comments:

2.1.1. dispersal is a function of distance and an activity-specific search weight. Does that imply mosquitoes always (if they survive) make it to another resource? i.e., movement is not random or undirected? Given the distances potentially involved in these landscapes, is that realistic?

P12. Launch and timing appears to be drawn randomly when the mosquito enters the resting phase. Does that imply that initiating host seeking doesn’t rely on the presence of hosts? E.g., a host entering a house wouldn’t activate opportunistic mosquito host seeking?

Landing on microsites: the model can accommodate a range of spatial scales, but if the goal indeed is to recreate realistic descriptions of local conditions, with the amount of detail that is required (e.g., all the microsites that comprise the locations where a mosquito could conceivably land and rest), it strikes me this might be most useful for tests at a small-scale. I would be interested in hearing the author’s thoughts (in the discussion) how one might actually go about using this model to investigate specific settings.

If sugar-feeding was not included in the current investigation (though energetic reserves were), the implications for movement (mortality incurred while foraging or traversing the distance between separated hosts and oviposition sites) might be drastically overestimated. How was this addressed?

Why do you include two distinct models of oogenesis (egg batch size as a draw from a distribution and as a function of blood meal size)? Is the idea that one of these captures a process of eggs entering a state of arrest with insufficient blood versus a process whereby primary follicles are resorbed in the case of a small blood meal? A clearer link to the underlying biological process might be helpful in understanding this.

L569: I’m having some trouble understanding what you mean by the mean value of R_2 being the average duration of one feeding cycle. Doesn’t this just give the proportion of mosquitoes that completes one cycle? Is this the mean time point at which mosquitoes entered the absorbing state?

Mapping the models onto each other: perhaps I’m missing it, but the equations presented for MBDETES don’t appear to allow for movement between sites, yet you mention testing this for three sites. Could you present the equations with this spatial component?

Fig 4: what does it imply that the egg laying rate is ~12 days? Is that the average time it takes before a mosquito lays a batch of eggs in these simulations (if so, that seems rather long)? What then is the feeding cycle duration of 1.2 days? And what is blood feeding by age, the time until mosquitoes on average take their first blood meal, or the rate at which they feed by age? For that matter, what is the time step used in the simulations? A bit more detail on these results would be helpful.

A final thought is that the results point to the importance of searching and foraging behavior that other models have found previously (e.g., the Saul paper that is cited). But other models have come to somewhat similar insights, e.g., Gu et al 2006, or Killeen & Smith 2007, and perhaps others. It would be interesting to see a comparison of these outcomes to those earlier ones, and whether there is anything about mosquito foraging that this more detailed model highlights.

Reviewer #2: These authors have developed two interacting models for describing mosquito behavior and movement. Variables within this model can be adjusted or turned on or off and there are many different behavior categories to try to encompass the different types of activities and decisions that mosquitoes engage in. This manuscript is written in a very clear and approachable way. It is one of the better model papers that I have read. The models also have very intuitive naming for mosquito behavioral state transitions.

The benefit of using a fine-scale model that is adjustable is that users can look at the impact of variables that they are most interested in and the model can be improved as we learn more about real-world parameters for mosquito behaviors across different species and environments. This model is not tuned to any disease vector particularly, but rather seeks to better describe the mosquito ecology of any obligatory bloodfeeding mosquito species. As such, it could be used to model mosquito dynamics in relation to disease control interventions and across a variety of disease-vectoring mosquitoes. One thing that some very fine-grain models suffer from is a lack of flexibility due to strict adherence to the parameters of what is known or assumed for one particular species. The flexibility of this set of models is their core strength, as it will enable the incorporation of better data as it becomes available. Additionally, a model allowing for heterogeneity in individual mosquitoes and modelling both male and female mosquitoes is really unique.

These authors appreciate the complexity of describing mosquito behaviors and the potential benefit of building that complexity into a tool. While not claiming to be predictive of real-life events, this is valuable for exploring how minor changes in mosquito biology and behavior can impact mosquito ecology and the transmission of pathogens.

Minor Comments:

The authors should make clear what is a general description of mosquito behavior as a form of introduction and what behaviors are included in the model.

Generally, what constitutes a “haunt” could be better defined in the text compared to what is a microenvironment or a resource landscape. The spatial aspects of this model do not come across as clearly as the other parameters described.

Some minor spelling errors throughout, just needs a bit of copyediting, particularly in the figure legends.

Figure 1 – the font on this figure is hard to read (and some of the wording is very small). Suggest consistent font and size across flow diagrams. The font in figures 2 and 3 is more legible. Minor spelling errors in the caption for figure 1.

Figures 7-10 are really interesting, and the discussion of them is pretty limited in the text. Seemingly, there is a lot of set up and explanation but not as much discussion of these outputs and their potential impact on a system. This section could be more balanced.

Line 244 – “from time to time” here, regarding sugar feeding behavior, is too general. Perhaps something like “at least once per day”, as this is the most consistent and frequent mosquito behavior. It would be worth adding a sentence that male and female mosquitoes engage in this behavior.

Line 301 – correct to “makes it possible to configure all of these options to consider a biological process of interest”

Line 391 – “digestion” may be oversimplify what is happing at this time – since digestion of a bloodmeal occurs over several days, suggest “for diuresis and the early stages of digestion to occur”

Line 490 – it would be worth writing out “vectorial capacity” in the heading here

**Have all data underlying the figures and results presented in the manuscript been provided?**

Reviewer #1: Yes

Reviewer #2: Yes

PLOS authors have the option to publish the peer review history of their article (what does this mean?). If published, this will include your full peer review and any attached files.

Reviewer #1: No

Reviewer #2: No

---

## [Decision Letter · Decision Letter 1]

21 Mar 2020

Dear Professor Smith,

We are pleased to inform you that your manuscript 'Vector bionomics and vectorial capacity as emergent properties of mosquito behaviors and ecology' has been provisionally accepted for publication in PLOS Computational Biology.

Best regards,

James Lloyd-Smith

Associate Editor

PLOS Computational Biology

Virginia Pitzer

Deputy Editor

PLOS Computational Biology

Reviewer's Responses to Questions

**Comments to the Authors:**

Reviewer #1: I commend the authors on their thorough and thoughtful point-by-point responses and the clarifications made to the manuscript. I'd be excited to see this accepted in PLoS CB.

**Have all data underlying the figures and results presented in the manuscript been provided?**

Reviewer #1: None

PLOS authors have the option to publish the peer review history of their article (what does this mean?). If published, this will include your full peer review and any attached files.

Reviewer #1: No

---

## [Editor Report · Acceptance letter]

6 Apr 2020

PCOMPBIOL-D-19-01652R1 

Vector bionomics and vectorial capacity as emergent properties of mosquito behaviors and ecology

Dear Dr Smith,

I am pleased to inform you that your manuscript has been formally accepted for publication in PLOS Computational Biology. Your manuscript is now with our production department and you will be notified of the publication date in due course.

With kind regards,

Laura Mallard
